# Kernel methods and their derivatives: Concept and perspectives for the earth system sciences

**J. Emmanuel Johnson** *, **Valero Laparra, Adrián Pérez-Suay** , **Miguel D. Mahecha**¤, **Gustau Camps-Valls**

Image Processing Laboratory, Universitat de València, València, Spain

¤ Current address: Max Planck Institute for Biogeochemistry, Jena, Germany
* juan.johnson@uv.es

## Abstract

Kernel methods are powerful machine learning techniques which use generic non-linear functions to solve complex tasks. They have a solid mathematical foundation and exhibit excellent performance in practice. However, kernel machines are still considered black-box models as the kernel feature mapping cannot be accessed directly thus making the kernels difficult to interpret. The aim of this work is to show that it is indeed possible to interpret the functions learned by various kernel methods as they can be intuitive despite their complexity. Specifically, we show that derivatives of these functions have a simple mathematical formulation, are easy to compute, and can be applied to various problems. The model function derivatives in kernel machines is proportional to the kernel function derivative and we provide the explicit analytic form of the first and second derivatives of the most common kernel functions with regard to the inputs as well as generic formulas to compute higher order derivatives. We use them to analyze the most used supervised and unsupervised kernel learning methods: Gaussian Processes for regression, Support Vector Machines for classification, Kernel Entropy Component Analysis for density estimation, and the Hilbert-Schmidt Independence Criterion for estimating the dependency between random variables. For all cases we expressed the derivative of the learned function as a linear combination of the kernel function derivative. Moreover we provide intuitive explanations through illustrative toy examples and show how these same kernel methods can be applied to applications in the context of spatio-temporal Earth system data cubes. This work reflects on the observation that function derivatives may play a crucial role in kernel methods analysis and understanding.

## 1 Introduction

Kernel methods (KMs) constitute a standard set of tools in machine learning and pattern analysis [1, 2]. They are based on a mathematical framework to cope with nonlinear problems while still relying on well-established concepts of linear algebra. KMs are one of the preferred

**Data Availability Statement:** All toy example code is reproducible and is available at: https://github.com/IPL-UV/sakame All applied data is open

source and available at: https://www.
earthsystemdatalab.net/.

**Funding:** GCV 647423 European Research Council
https://erc.europa.eu/ The funders had no role in
study design, data collection and analysis, decision
to publish, or preparation of the manuscript.

**Competing interests:** The authors have declared
that no competing interests exist.

tools in applied sciences, from signal and image processing [3], to computer vision [4] and geosciences [5]. Since its introduction in the 1990s through the popular support vector machines (SVMs), kernel methods have evolved into a large family of techniques that cope with many problems in addition to classification. Kernel machines have also excelled in regression, interpolation and function approximation problems [3], where Gaussian Processes (GPs) [6] and support vector regression [7] have provided good results in many applications. Furthermore, many kernel methods have been engineered to deal with other relevant learning problems; for example, density estimation via kernel decompositions using entropy components [8]. For dimensionality reduction and feature extraction, there are a wide family of multivariate data analysis kernel methods such as kernel principal component analysis [9], kernel canonical analysis [10] or kernel partial least squares [11]. Kernels have also been exploited to estimate dependence (nonlinear associations) between random variables such as kernel mutual information [12], or the Hilbert-Schmidt Independence Criterion [13]. Finally in the literature, we find kernel machines for data sorting [14], manifold learning and alignment [15], system identification [16], signal deconvolution and blind source separation [3].

However, *understanding* a model is more difficult than just *applying* a model, and kernel methods are still considered black-box models. Little can be said about the characteristics of the feature mapping which is only implicit in the formulation. Several approaches have been presented in the literature to *explore* the kernel feature mapping and to understand what the kernel machine is actually learning. One way to analyze kernel machines is by visualizing the empirical feature maps but this is very challenging and only feasible in low-dimensional problems [1, 17]. Another approach is to study the relative relevance of the input features (covariates) on the output. This is commonly referred to as feature ranking and it typically reduces to evaluating how the function varies when an input is removed or perturbed. Automatic relevance determination (ARD) kernels [6] or multiple kernel learning [18] allow one to study the relevance of the feature components indirectly. While this approach has been extensively used to improve the accuracy and understanding of supervised kernel classifiers and regression methods, they only provide feature ranking and nothing is said about the geometrical properties of the feature map. In order to resolve this, two main approaches are available in the kernel methods literature. For some particular kernels one can derive the metric induced by the kernel to give insight into the surfaces and structures [19]. Alternatively, one can study the feature map (in physically meaningful units) by learning the inverse feature mapping; a group of techniques known as kernel pre-imaging [20, 21]. However, the current methods are computationally expensive, involve critical parameters, and very often provide unstable results.

Function derivatives is a classical way to describe and visualize some characteristics of models. Derivatives of kernel functions have been introduced before, yet mostly used in supervised learning as a form of regularization that controls fast variations of the decision function [22]. However, derivatives of the model's function with regards to the input features for feature understanding and visualization has received less attention. A recent strategy is to derive sensitivity maps from a kernel feature map [23]. The sensitivity map is related to the squared derivative of the function with respect to the input features. The idea was originally derived for SVMs in neuroimaging applications [24], and later extended to GPs in geoscience problems [25–28]. In both cases, the goal was to retrieve a feature ranking from a learned supervised model.

In this paper, we analyze the kernel function derivatives for supervised and unsupervised kernel methods with several kernel functions in different machine learning paradigms. We show the usefulness of the derivatives to study and visualize kernel models in regression, classification, density estimation, and dependence estimation with kernels. Since differentiation is a linear operator, most kernel methods have a derivative that is proportional to the derivative of

the kernel function. We provide the analytic form of the first and second derivatives of the most common kernel functions with regards to the inputs, along with iterative formulas to compute the $m$-th order derivative of differentiable kernels, and for the radial basis function kernel in particular; where $m$ is the number of successive derivatives. In classification problems, the derivatives can be related to the margin, and allow us to gain some insight on sampling [29]. In regression problems, a models' function derivatives may give insight about the signal and noise characteristics that allow one to design regularization functionals. In density estimation, the second derivative (the Hessian) allows us to follow the density ridge for manifold learning [30], whereas in dependence estimation squared derivatives (the sensitivity maps) allows one to study the most relevant points and features governing the association measure [31]. All in all, kernel derivatives allow us to identify both examples and features that affect the predictive function the most, and allow us to interpret the kernel model behavior in different learning applications. We show that the solutions can be expressed in closed-form for the most common kernel functions and kernel methods, they are easy to compute, and we give examples of how they can be used in practice.

The remainder of the paper is organized as follows. Section 2 briefly reviews the fundamentals of kernel functions and feature maps, and concentrates on the kernel derivatives for feature map analysis where we provide the first and second order derivatives for most of the common kernel functions. We also review the main ideas to summarize the information contained in the derivatives. Section 3 and Section 4 study popular discriminative kernel methods, such as Gaussian Processes for regression and support vector machines for classification. Section 5 analyzes the interesting case of density estimation with kernels, in particular through the use of kernel entropy component analysis for density estimation. Section 6 pays attention to the case of dependence estimation between random variables using the Hilbert-Schmidt independence criterion in cases of dependence visualization maps and data unfolding. Section 7 illustrates the applicability of kernel derivatives in the previous kernel methods on spatio-temporal Earth system science data. We conclude in section 8 with some final remarks.

## 2 Kernel functions and the derivatives

### 2.1 Kernel functions and feature maps

In this section, we briefly highlight the most important properties of kernel methods, needed to understand their role of the kernel methods mentioned in the subsequent sections. Recall that kernel methods rely on the notion of similarity between points in a higher (possibly infinite) dimensional Hilbert space. Let us consider a set of empirical data $\mathcal{X} = \{\boldsymbol{x}_1, \ldots, \boldsymbol{x}_n\}$, whose elements are defined in a $d$-dimensional input space, $\boldsymbol{x}_i = [x_i^1, \ldots, x_i^d]^\top \in \mathbb{R}^d, 1 \le i \le n$. In supervised settings, each input feature vector $\boldsymbol{x}$ is associated with a target value, which can be either discrete in the classification case, $y_i \in \mathbb{Z}^+$ or real in the regression case, $y_i \in \mathbb{R}, i = 1, \ldots, n$. Kernel methods assume the existence of a Hilbert space $\mathcal{H}$ with an inner product $\langle \cdot, \cdot \rangle_{\mathcal{H}}$ where samples in $\mathcal{X}$ are mapped into with a feature map $\phi : \mathcal{X} \to \mathcal{H}, \boldsymbol{x}_i \mapsto \phi(\boldsymbol{x}_i), 1 \le i \le n$. The mapping function can be defined explicitly (if some prior knowledge about the problem is available) or implicitly, which is often the case in kernel methods. The similarity between the elements in $\mathcal{H}$ can be estimated using its associated dot product $\langle \cdot, \cdot \rangle_{\mathcal{H}}$ via reproducing kernels in Hilbert spaces (RKHS), $k : \mathcal{X} \times \mathcal{X} \to \mathbb{R}$, such that pairs of points $(\boldsymbol{x}, \boldsymbol{x}') \mapsto k(\boldsymbol{x}, \boldsymbol{x}')$. So we can estimate similarities in $\mathcal{H}$ without the explicit definition of the *feature map $\varphi$*, and hence without having access to the points in $\mathcal{H}$. This *kernel function $k$* is required to satisfy Mercer's Theorem [32].

**Definition 1** *Reproducing kernel Hilbert spaces (RKHS)* [33]. *A Hilbert space $\mathcal{H}$ is said to be a RKHS if: (1) The elements of $\mathcal{H}$ are complex or real valued functions $f(\cdot)$ defined on any set of elements $\boldsymbol{x}$; And (2) for every element $\boldsymbol{x}$, $f(\cdot)$ is bounded.*

The name of these spaces comes from the so-called *reproducing property*. In a RKHS $\mathcal{H}$, there exists a function $k(\cdot, \cdot)$ such that

$$f(\boldsymbol{x}) = \langle f, k(\cdot, \boldsymbol{x}) \rangle_{\mathcal{H}}, \quad f \in \mathcal{H}, \tag{1}$$

by virtue of the Riesz Representation Theorem [34]. In particular, for any $\boldsymbol{x}, \boldsymbol{x}' \in \mathcal{X}$

$$k(\boldsymbol{x}, \boldsymbol{x}') = \langle k(\cdot, \boldsymbol{x}), k(\cdot, \boldsymbol{x}') \rangle_{\mathcal{H}} \tag{2}$$

A large class of algorithms have originated from regularization schemes in RKHS. The *representer theorem* gives us the general form of the solution to the common loss function formed by the loss term and a regularization term.

**Theorem 1** (*Representer Theorem*) [34, 35] *Let $\Omega : [0, \infty) \to \mathbb{R}$ be a strictly monotonic increasing function; let $\boldsymbol{V} : (\mathcal{X} \times \mathbb{R}^2)^n \to \mathbb{R} \cup \{\infty\}$ be an arbitrary loss function; and let $\mathcal{H}$ be a RKHS with reproducing kernel k. Then*:

$$f^* = \min_{f \in \mathcal{H}} \{ \boldsymbol{V}((\boldsymbol{x}_1, y_1, f(\boldsymbol{x}_1)), \dots, (\boldsymbol{x}_n, y_n, f(\boldsymbol{x}_n))) + \Omega(\| f \|_{\mathcal{H}}^2) \} \tag{3}$$

*admits a space of functions f defined as*

$$f(\boldsymbol{x}) = \sum_{i=1}^{n} \alpha_i k(\boldsymbol{x}, \boldsymbol{x}_i), \quad \alpha_i \in \mathbb{R}, \quad \alpha \in \mathbb{R}^n, \tag{4}$$

which is expressed as a linear combination of kernel functions. Also note that the previous theorem states that solutions imply having access to an empirical risk term **V** and a regularizer $\Omega$. In the case of not having labels $y_i$, alternative representer theorems can be equally defined. A generalized representer theorem was introduced in [36], which generalizes Wahba's theorem to a larger class of regularizers and empirical losses. Also, in [37], a representer theorem for kernel principal components analysis (KPCA) was used: the theorem gives the solution as a linear combination of kernel functions centered at the input data points, and is called the representer theorem of learning theory [38], whereby the coefficients are determined by the eigendecomposition of the kernel matrix [9, 36]. Should the reader want more literature related to kernel methods, we highly recommend this paper [39] for a more theoretical introduction to Hilbert-Spaces in the context of kernel methods and [3] for a more applied and practical approaches.

## 2.2 Derivatives of linear expansions of kernel functions

Computing the derivatives of function *f* can give important insights about the learned model. Interestingly, in the majority of kernel methods, the function *f* is linear in the parameters $\boldsymbol{\alpha}$, cf. Eq (4) derived from the representer theorem [35] [Th. 1]. For the sake of simplicity, we will denote the partial derivative of *f* w.r.t. the feature $x^j$ as $\partial_j f(\boldsymbol{x}) = \frac{\partial f(\boldsymbol{x})}{\partial x^j}$, where *j* denotes the dimension. This allows us to write the partial derivative of *f* as:

$$\partial_j f(\boldsymbol{x}) := \frac{\partial f(\boldsymbol{x})}{\partial x^j} = \frac{\partial \sum_{i=1}^{n} \alpha_i k(\boldsymbol{x}, \boldsymbol{x}_i)}{\partial x^j} = \sum_{i=1}^{n} \alpha_i \frac{\partial k(\boldsymbol{x}, \boldsymbol{x}_i)}{\partial x^j} = (\partial_j \boldsymbol{k}(\boldsymbol{x}))^\top \boldsymbol{\alpha}, \tag{5}$$

where $\partial_j \boldsymbol{k}(\boldsymbol{x}) := \left[ \frac{\partial k(\boldsymbol{x}, \boldsymbol{x}_1)}{\partial x^j}, \dots, \frac{\partial k(\boldsymbol{x}, \boldsymbol{x}_n)}{\partial x^j} \right]^\top \in \mathbb{R}^n$ and $\boldsymbol{\alpha} = [\alpha_1, \dots, \alpha_n]^\top \in \mathbb{R}^n$. It is possible to take

the second order derivative with respect to feature $x^j$ twice which remains linear as well with $\boldsymbol{\alpha}$:

$$\partial_j^2 f(\boldsymbol{x}) := \frac{\partial^2 f(\boldsymbol{x})}{\partial x^j \partial x^j} = \frac{\partial^2 \sum_i \alpha_i k(\boldsymbol{x}, \boldsymbol{x}_i)}{\partial x^{j2}} = \sum_i \alpha_i \frac{\partial^2 k(\boldsymbol{x}, \boldsymbol{x}_i)}{\partial x^{j2}} = (\partial_j^2 \boldsymbol{k}(\boldsymbol{x}))^\top \boldsymbol{\alpha}, \tag{6}$$

where $\partial_j^2 \boldsymbol{k}(\boldsymbol{x}) := \left[ \frac{\partial^2 k(\boldsymbol{x}, \boldsymbol{x}_1)}{\partial x^{j2}}, \ldots, \frac{\partial^2 k(\boldsymbol{x}, \boldsymbol{x}_n)}{\partial x^{j2}} \right]^\top \in \mathbb{R}^n$. Inductively, the $m$-th partial derivative w.r.t the $j$-th feature is also linear with $\boldsymbol{\alpha}$ and it follows the following equation:

$$\partial_j^m f(\boldsymbol{x}) = (\partial_j^m \boldsymbol{k}(\boldsymbol{x}))^\top \boldsymbol{\alpha}. \tag{7}$$

The gradient of $f$ gives information about the slope (increase rate) of the function and reduces to

$$\nabla f = \left[ \frac{\partial f(\boldsymbol{x})}{\partial x^1}, \ldots, \frac{\partial f(\boldsymbol{x})}{\partial x^d} \right]^\top = (\nabla \mathbf{K})^\top \boldsymbol{\alpha} \in \mathbb{R}^d, \tag{8}$$

where $\nabla$ denotes the vector differential operator, and $\nabla \mathbf{K} = [\partial_1 \boldsymbol{k}(\boldsymbol{x}) | \cdots | \partial_d \boldsymbol{k}(\boldsymbol{x})]$. The Laplacian accounts for the curvature, roughness, or concavity of the function itself, and can be easily computed as the sum of all the unmixed second partial derivatives, which for kernels reduces to

$$\nabla^2 f := \sum_{j=1}^d \frac{\partial^2 f(\boldsymbol{x})}{\partial x^{j2}} = \mathbb{1}_d^\top (\nabla^2 \mathbf{K})^\top \boldsymbol{\alpha} \in \mathbb{R}, \tag{9}$$

where $\nabla^2 \mathbf{K} = [\partial_1^2 \boldsymbol{k}(\boldsymbol{x}) | \cdots | \partial_d^2 \boldsymbol{k}(\boldsymbol{x})]$ and $\mathbb{1}_d$ is a column vector of ones of size $d$. Another useful descriptor is the Hessian matrix of $f$, which characterizes its local curvature. The Hessian is a $d \times d$ matrix of second-order partial derivatives with respect to the features $x^j$, $x^k$:

$$[\boldsymbol{H}]_{jk} = \frac{\partial^2 f(\boldsymbol{x})}{\partial x^j \partial x^k} = (\partial_j \partial_k \boldsymbol{k}(\boldsymbol{x}))^\top \boldsymbol{\alpha} \in \mathbb{R}. \tag{10}$$

The equations listed above have shown that the derivative of a kernel function is linear with $\boldsymbol{\alpha}$. Once the $\boldsymbol{\alpha}$ is computed, the problem reduces to (1) computing the derivatives for a particular kernel function, and (2) to summarize the information contained within the derivatives.

- **Derivatives of common kernel functions**. Kernel methods typically use a set of positive definite kernel functions, such as the linear, polynomial (Poly), hyperbolic tangent (Tanh), Gaussian (RBF) kernel, and the automatic relevance determination (ARD) kernel. We give the partial derivative for all of these kernels in Table 1, and the (mixed) second derivatives in Table 2. For the most widely used kernels (RBF and ARD), one can recognize a linear relation between the kernel derivative and the kernel function itself. It can be shown that the $m$-

**Table 1. Partial derivatives for some common kernel functions: Linear, Polynomial (Poly), Radial Basis Functions (RBF), Hyperbolic tangent (Tanh), and Automatic Relevance Determination (ARD).**

| Kernel | Kernel function, $k(x, y)$ | Partial derivative, $\frac{\partial k(x,y)}{\partial x^j}$ |
|---|---|---|
| Linear | $\boldsymbol{x}^\top \boldsymbol{y}$ | $y^j$ |
| Poly | $(\gamma \boldsymbol{x}^\top \boldsymbol{y} + c_0)^p$ | $\gamma p y^j (\gamma \boldsymbol{x}^\top \boldsymbol{y} + c_0)^{p-1}$ |
| RBF | $\exp(-\gamma \|\boldsymbol{x} - \boldsymbol{y}\|^2)$ | $-2\gamma(x^j - y^j)k(\boldsymbol{x}, \boldsymbol{y})$ |
| Tanh | $\tanh(\gamma \boldsymbol{x}^\top \boldsymbol{y} + c_0)$ | $\gamma y^j \operatorname{sech}^2(\gamma \boldsymbol{x}^\top \boldsymbol{y} + c_0)$ |
| ARD | $v^2 \exp\left( -\frac{1}{2} \sum_{d=1}^D \left( \frac{x^d - y^d}{\lambda_d} \right)^2 \right)$ | $\left( \frac{x^j - y^j}{\lambda_j^2} \right) k(\boldsymbol{x}, \boldsymbol{y})$ |

**Table 2. Second derivatives for some common kernel functions.**

| Kernel | 2nd partial derivative, $\frac{\partial^2 k(x,y)}{\partial x^{j^2}}$ | Mixed partial derivative, $\frac{\partial^2 k(x,y)}{\partial x^j \partial x^k}$ |
|---|---|---|
| Linear | 0 | 0 |
| Poly | $(p-1)p\,(\gamma y^j)^2\,(\gamma\,\boldsymbol{x}^\top\boldsymbol{y}+c_0)^{p-2}$ | $(p-1)p\gamma^2\,y^j\,y^k\,(\gamma\,\boldsymbol{x}^\top\boldsymbol{y}+c_0)^{p-2}$ |
| RBF | $2\gamma\,[2\gamma\,(x^j-y^j)^2-1]\,k(\boldsymbol{x},\boldsymbol{y})$ | $4\gamma^2\,(x^k-y^k)\,(x^j-y^j)\,k(\boldsymbol{x},\boldsymbol{y})$ |
| Tanh | $-2(\gamma y^j)^2\,\mathrm{sech}^2(\gamma\,\boldsymbol{x}^\top\boldsymbol{y}+c_0)k(\boldsymbol{x},\boldsymbol{y})$ | $-2\gamma^2\,y^j\,y^k\,\mathrm{sech}^2(\gamma\,\boldsymbol{x}^\top\boldsymbol{y}+c_0)k(\boldsymbol{x},\boldsymbol{y})$ |
| ARD | $\left(\frac{1}{\lambda_j^2}+\left(\frac{x^j-y^j}{\lambda_j^2}\right)^2\right)k(\boldsymbol{x},\boldsymbol{y})$ | $\left(\frac{x^j-y^j}{\lambda_j^2}\right)\left(\frac{x^k-y^k}{\lambda_k^2}\right)k(\boldsymbol{x},\boldsymbol{y})$ |

th derivative of some kernel functions can be computed recursively using Faà di Bruno's identity [40].

- **Summarizing function derivatives**. Summarizing the information contained in the derivatives is not an easy task, especially in high dimensional problems. The most obvious strategy is to use the norm of the partial derivative, that is $\|\partial_j f\|$, which summarizes the relevance of variable $x^j$. A small norm implies a small change in the discriminative function $f$ with respect to the $j$-th dimension, indicating the low importance of that feature. This approach was introduced as *sensitivity maps* (SMs) in [24] for the visualization of SVM maps in neuroimaging and later exploited in GPs for ranking spectral channels in geosciences applications [26]. The SM for the $j$-th feature, is the expected value of the squared derivative of the function with respect the input argument $x^j$:

$$s^j = \int_{\mathcal{X}^j}\left(\frac{\partial f(\boldsymbol{x})}{\partial x^j}\right)^2 p(x^j)\mathrm{d}x^j, \tag{11}$$

where $p(x)$ is the probability density function (pdf) over dimension $j$ of the input space $\mathcal{X}$. In order to avoid the possibility of cancellation of the terms due to its signs, the derivatives are squared. Other transformations like the absolute value could be equally applied. The *empirical sensitivity map* approximation to Eq (11) is obtained by replacing the expected value with a summation over the available $n$ samples

$$s^j \approx \frac{1}{n}\sum_{i=1}^{n}\left(\frac{\partial f(\boldsymbol{x}_i)}{\partial x_i^j}\right)^2, \tag{12}$$

which can be grouped together to define the *sensitivity vector* as $\mathbf{s} = [s^1, \ldots, s^d]^\top$.

This can be thought of as studying the relevance of the sample points. Similarly, one can average over the features to obtain a *point sensitivity*:

$$q_i = \frac{1}{d}\sum_{j=1}^{d}\left(\frac{\partial f(\boldsymbol{x}_i)}{\partial x_i^j}\right)^2, \tag{13}$$

which can be grouped to define the *point sensitivity vector* as $\mathbf{q} = [q_1, \ldots, q_n]^\top$. The information contained in $\mathbf{q}$ is related to the robustness to changes of the decision in each point of the space.

Now we are equipped to use the derivatives and the corresponding sensitivity maps in arbitrary kernel machines that use standard kernel functions. In the following sections, we study

its use in kernel methods for both supervised (regression and classification) and unsupervised (density estimation and dependence estimation) learning.

## 3 Kernel regression

### 3.1 Gaussian Process Regression

Multiple proposals to use kernel methods in a regression framework have been done during the last few decades. Gaussian Processes (GPs) is perhaps the most successful kernel method for discriminative learning in general and regression in particular [6]. Standard GP regression approximates observations as the sum of some unknown latent function $f(\boldsymbol{x})$ of the inputs plus some additive Gaussian noise, $y_i = f(\boldsymbol{x}_i) + \varepsilon_i$, where $\varepsilon_i \sim \mathcal{N}(0, \sigma^2)$. A zero mean GP prior is placed on the latent function $f(\boldsymbol{x})$ and a Gaussian prior is used for each latent noise term $\varepsilon_i$, in other words $f(\boldsymbol{x}) \sim \mathcal{GP}(m(\boldsymbol{x}), \mathbf{K})$, where $m(\boldsymbol{x}) = 0$, and $\mathbf{K}$ is a covariance function, $[\mathbf{K}]_{ij} = k(\boldsymbol{x}_i, \boldsymbol{x}_j)$, parameterized by a set of hyperparameters $\boldsymbol{\theta}$ (e.g. $\boldsymbol{\theta} = [\lambda_1, \ldots, \lambda_d]$ for the ARD kernel function).

If we consider a test location $\boldsymbol{x}_*$ with the corresponding output $y_*$, a $\mathcal{GP}$ prior induces a prior distribution between the observations $\boldsymbol{y}$ and $y_*$. Collecting all available data in $\mathcal{D} \equiv \{(\boldsymbol{x}_i, y_i) | i = 1, \ldots n\}$, it is possible to analytically compute the posterior distribution over the unknown output $y_*$ given the test input $\boldsymbol{x}_*$ and the available training set $\mathcal{D}$, $p(y_* | \boldsymbol{x}_*, \mathcal{D}) = \mathcal{N}(y_* | \mu_{\mathrm{GP}*}, \sigma^2_{\mathrm{GP}*})$, which is a Gaussian with the following mean and variance:

$$\mu_{\mathrm{GP}*} = \boldsymbol{k}_*^\top (\mathbf{K} + \sigma_n^2 \mathbf{I})^{-1} \boldsymbol{y} = \boldsymbol{k}_*^\top \boldsymbol{\alpha}, \qquad (14)$$

$$\sigma^2_{\mathrm{GP}*} = \sigma_n^2 + k_{**} - \boldsymbol{k}_*^\top (\mathbf{K} + \sigma_n^2 \mathbf{I})^{-1} \boldsymbol{k}_*, \qquad (15)$$

where $\boldsymbol{k}_* = \boldsymbol{k}(\boldsymbol{x}_*) = [k(\boldsymbol{x}_*, \boldsymbol{x}_1), \ldots, k(\boldsymbol{x}_*, \boldsymbol{x}_n)]^\top \in \mathbb{R}^n$ contains the kernel similarities of the test point $\boldsymbol{x}_*$ to all training points in $\mathcal{D}$, $\mathbf{K}$ is a $n \times n$ kernel (covariance) matrix whose entries contain the similarities between all training points, $\boldsymbol{y} = [y_1, \ldots, y_n]^\top \in \mathbb{R}^n$, $k_{**} = k(\boldsymbol{x}_*, \boldsymbol{x}_*)$ is a scalar with the self-similarity of $\boldsymbol{x}_*$, and $\boldsymbol{I}$ is the identity matrix. The solution of the predictive mean for the GP model in (14) is expressed in the same way as equation (4), where $\mu_{\mathrm{GP}*} = f(\boldsymbol{x}_*) = \boldsymbol{k}_*^\top \boldsymbol{\alpha}$. This expression is exactly the same as in other kernel regression methods like the Kernel Ridge Regression (KRR) [2] or the Relevance Vector Machine (RVM) [2]. The derivative of the mean function can be computed through Eq (5) and the derivatives in Table 1.

### 3.2 Derivatives and sensitivity maps

Let us start by visualizing derivatives in simple 1D examples. We used GP modeling with a standard RBF kernel function to fit five regression data sets. We show in Fig 1 the first and second derivatives of the fitted GP model, as well as the point-wise sensitivities. In all cases, first derivatives are related to positive or negative slopes, while the second derivatives are related to the curvature of the function. Since the derivative is a linear operator, a composition of functions is also the composition of derivatives as can be seen in the last two functions. This could be useful for analyzing more complex composite kernels. See Table 3 for a comparison with other kernel methods derivatives.

### 3.3 Derivatives and regularization

We show an example of applying the derivative of the kernel function as a regularization parameter for the noise. We modeled the function $f(x) = \sin(3\pi x)$ with an additive white

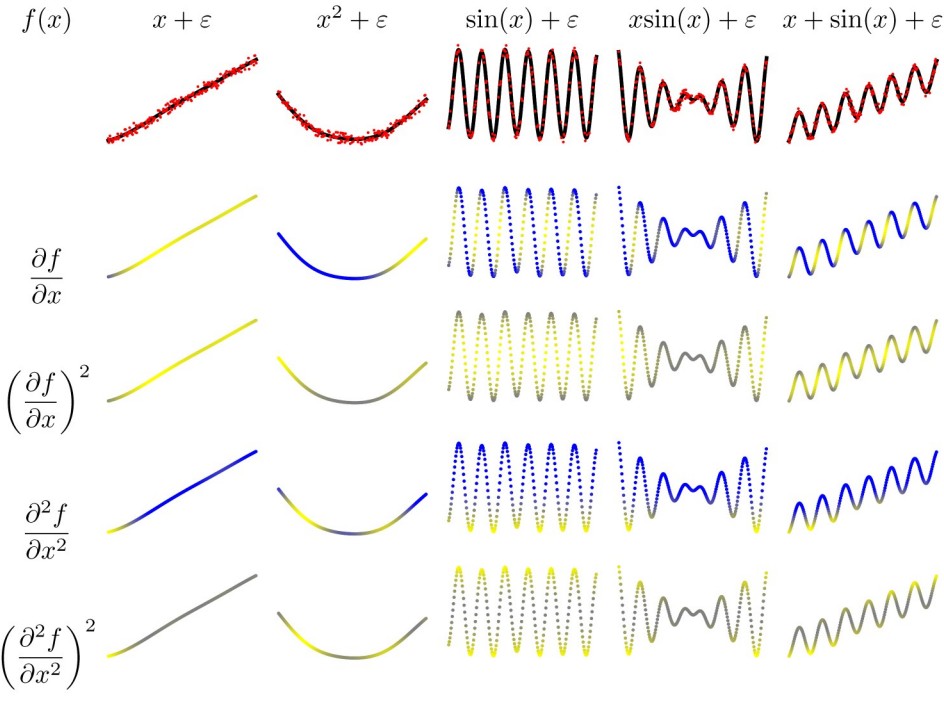

**Fig 1. Different examples of functions, derivatives and sensitivity maps.** Original data (red), the GP predictive function (black), high derivative values are in yellow, close-to-zero derivative values are in gray and negative derivative values are in blue.

Gaussian noise (AWGN) $n \sim \mathcal{N}(0, \sigma_n^2)$ using a Kernel Ridge Regression (KRR) model with RBF kernel. Different amounts of noise power $\sigma_n^2$ was used resulting in different values of the signal to noise ratio (SNR), SNR $= 10 \log(\sigma_y^2/\sigma_n^2)$, SNR$\in[0, 50]$ dB. Two different settings were explored to analyze the impact of the standard regularizer, $\| f \|_{\mathcal{H}}^2$, and the derivatives in KRR modeling: (1) either using the optimal amount of regularization in (14), $\sigma_n^2 = \sigma_r^2$, or (2) assuming no regularization was needed, $\sigma_n^2 = 0$.

Four scenarios were explored in this experiment: $\| f \|_{\mathcal{H}}^2 = \boldsymbol{\alpha}^\top \mathbf{K} \boldsymbol{\alpha}$, $\| f \|_2^2 = \boldsymbol{\alpha}^\top \mathbf{K}^\top \mathbf{K} \boldsymbol{\alpha}$, $\| \nabla f \|_2^2 = \boldsymbol{\alpha}^\top (\nabla \mathbf{K})^\top (\nabla \mathbf{K}) \boldsymbol{\alpha}$, and $\| \nabla^2 f \|_2^2 = \boldsymbol{\alpha}^\top (\nabla^2 \mathbf{K})^\top (\nabla^2 \mathbf{K}) \boldsymbol{\alpha}$, where $\mathbf{K}$ is a matrix with entries $[\mathbf{K}]_{ij} = k(\boldsymbol{x}_i, \boldsymbol{x}_j)$ (for definitions of gradients see Eqs (8) and (9)). The resulting SNR curves were then normalized in such a way that they are comparable. We explore two scenarios; the regularized and unregularized. Since the maximum SNR was subtracted from all norm values, in Fig 2a any norm greater than zero signifies the need to regularize more and in Fig 2b any norm less than zero signifies the need to regularize less.

**Table 3. Summary of the formulation for each of the main kernel methods GPR (Gaussian Process Regression, section 3), SVM (Support Vector Machines, section 4), KDE (Kernel Density Estimation, section 5), HSIC (Hilbert-Schmidt Independence Criterion, section 6).** The derivative formulation as well as some related analysis procedures in the literature as well as demonstrated in this paper.

| Method | Function | Derivative | Analysis |
|---|---|---|---|
| GPR | $\mathbf{k}_*^\top \alpha$ | $\partial \mathbf{k}_*^\top \alpha$ | Sensitivity, Ranking, Regularization |
| SVM | $g(y\alpha \mathbf{k}_* + b)$ | $(1 - g^2(\boldsymbol{x}_*)) \partial \mathbf{k}_*^\top y \alpha$ | Sensitivity, Feature Ranking, Margin |
| KDE | $n^{-1}\boldsymbol{k}_* \mathbb{1}_n$ | $\nabla \hat{p}(\boldsymbol{x}_*)^\top \mathbf{E}_r(\boldsymbol{x}_*)$ | Principal Curves |
| HSIC | $n^{-2}\mathrm{Tr}(\mathbf{KHLH})$ | $2n^{-2} \mathbf{A}_i \partial_q \boldsymbol{k}(\mathbf{x}_i)$ | Leverage, Feature/Point Relevance |

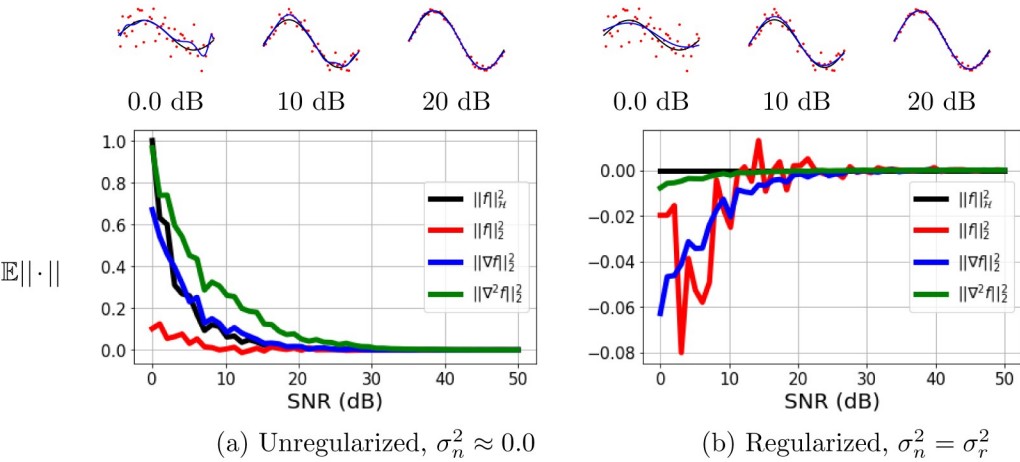

(a) Unregularized, $\sigma_n^2 \approx 0.0$ (b) Regularized, $\sigma_n^2 = \sigma_r^2$

**Fig 2. Signal-to-Noise Ratio (SNR) versus the expected normalized value of different norms ($\mathbb{E}||\cdot||$) to act as regularizers.** A unregularized (left) and an regularized (right) Kernel Ridge Regression (KRR) model was fitted. The top row shows a few examples of these fitted KRR models with a different quantity of noise added. The red data points are the data with different noise levels, the true function is black and the fitted KRR model is in blue. The second row shows the norm for the different regularizers. All lines were normalized in such a way that they are comparable. The norm of the true signal (SNR = 50 dB) is subtracted from all points so any curve with values below zero require less regularization and any points above zero require more regularization.

Fig 2 shows the effect of the noise on the norm for different regularization terms. All four regularization functions give the user information about how noisy the signal is for the unregularized case and the regularized case. The graph for the regularized case has the norms of the functions below zero, except for the $\| f \|_{\mathcal{H}}^2$, when the SNR is extremely low. Since the norm of the functions are increasing as one increases the SNR, this says that there needs to be less regularization. The $\| f \|_{\mathcal{H}}^2$ has a straight line because the 'optimal' parameter for using the norm of the weights for regularization has already been chosen. However, the norm of the first and second derivative still give us information that the problem needs to be regularized less. So both cases showcase the functionality of the first and second derivative as viable regularizers.

## 4 Kernel classification

### 4.1 Support vector machine classification

The first effective and influential kernel method introduced was the Support Vector Machine (SVM) [1, 41–43] classifier. Researchers and practitioners have used it to solve problems in speech recognition [44], computer vision and image processing [45–47], or channel equalization [48]. The binary SVM classification algorithm minimizes a weighted sum of a loss and a regularizer

$$\sum_{i=1}^{n} \mathbf{V}(y_i, f(\boldsymbol{x}_i)) + \lambda \| f \|_{\mathcal{H}_k}^2,$$

where the cost function is called the 'hinge loss' and is defined as
$\mathbf{V}(y_i, f(\boldsymbol{x}_i)) = \max(0, 1 - y_i \hat{f}(\boldsymbol{x}_i))$, $\boldsymbol{y}_i \in \{-1, +1\}$, $f \in \mathcal{H}_k$ and $\mathcal{H}_k$ is the RKHS of functions generated by the kernel $k$, and $\lambda$ is a parameter that trades off accuracy for smoothness. The norm $\| f \|_{\mathcal{H}_K}$ is generally interpreted as a roughness penalty, and can be expressed as a

function of kernels, $\| f \|_{\mathcal{H}_K} = f^\top K f$. The decision function for any test point $\boldsymbol{x}_*$ is given by

$$\hat{y}_* = g(f(\boldsymbol{x})) = \operatorname{sgn}\left(\sum_{i=1}^n y_i \alpha_i k(\boldsymbol{x}_*, \boldsymbol{x}_i) + b\right), \tag{16}$$

where $\alpha_i$ are Lagrange multipliers obtained from solving a quadratic programming (QP) problem, being the *support vectors* (SVs) of those training samples $\boldsymbol{x}_i$ with non-zero Lagrange multipliers $\alpha_i \neq 0$ [1]. See [49] for more details on the formulation and more practical examples.

## 4.2 Function derivatives and margin

The SVM decision function in (16) uses a mask function $g(x) = \operatorname{sgn}(\cdot)$ to decide between the two classes, which is inherited from the hinge loss used. Since the $\operatorname{sgn}(\cdot)$ function is not differentiable at 0 and for the sake of analytic tractability we replaced it with the hyperbolic tangent, $g(\cdot) = \tanh(\cdot)$. Now one can simply compute the derivative of the model by applying the chain rule:

$$\frac{\partial g(\boldsymbol{x}_*)}{\partial x_*^j} = \frac{\partial g(\boldsymbol{x}_*)}{\partial f(\boldsymbol{x}_*)}\frac{\partial f(\boldsymbol{x}_*)}{\partial x_*^j} = (1 - g^2(\boldsymbol{x}))\frac{\partial f(\boldsymbol{x}_*)}{\partial x_*^j} \tag{17}$$

where the leftmost term in the product can be seen as a mask function on top of the derivative of the regression function and allows us to study the model in terms of decision and estimation separately. See Table 3 for a comparison to other kernel methods derivatives.

Three datasets were used to illustrate the effect of the derivative in the SVM classifier. We used a SVM with RBF kernel in all cases, and hyperparameters were tuned by 3-fold cross-validation and the results are displayed in Fig 3. The mask function only focuses on regions along the decision boundary. However the derivative of the kernel function displays a few regions along the decision boundary along with other regions outside of the decision boundary. The composite of the derivative of the masking function and kernel function showcases a combination of the two components: the high derivative regions along the decision boundary. The two half moons and two circles examples have a clear decision boundary and the derivative of the composite function is able to capture this. However, the two ellipsoid example is less clear as the decision boundary passes through two overlapping classes. This is related to the density within the margin as the regions with less samples have a smaller slope and the regions with more samples have a higher slope, which results in wider and thinner margin, respectively. This fact could be used to define more efficient sampling procedures.

## 5 Kernel density estimation

The problem of density estimation is difficult in machine learning and statistics and it has been widely studied via kernels [50–52]. Kernel density estimation (KDE) is a classical non-parametric method for estimating a probability density function (pdf) [53]. In KDE, the choice of the kernel function is key to properly approximating the underlying pdf from a finite number of samples. The KDE kernel must be a non-negative function that integrates to one (i.e. a proper pdf), yet does not need to be positive semi-definite (PSD). KDE is versatile in that sense. However, if the kernel is PSD, there are close relations between density estimation and RKHS learning via the kernel eigendecomposition. Many KDE kernels are PSD, and some well-known examples include the Gaussian kernel, the Student kernel and the Laplacian kernel [54] functions.

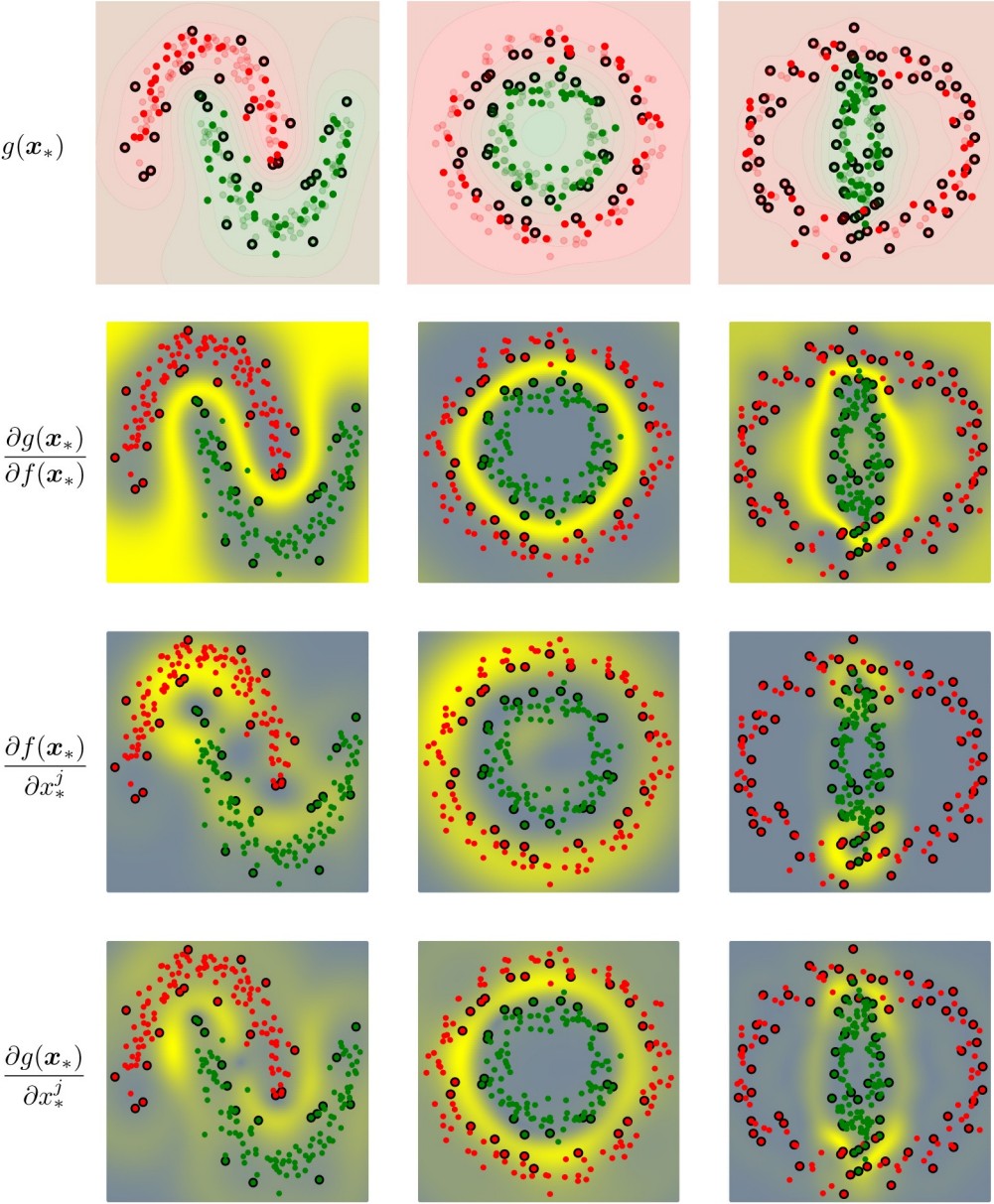

**Fig 3. Visualizing three examples of sensitivity maps in SVM classification.** The top row shows a figure has red and green points to showcase the classes, black points showing the support vectors chosen by the SVM classifier, and a contour map showcasing the same color scheme for the decision function. In the subsequent plots, we plot the sensitivity measures where the high derivative values are in yellow and negative derivative values are in gray. The leftmost column showcases which derivative is plotted.

## 5.1 Density estimation with kernels

For the Parzen window expression, KDE defines the pdf as a sum of kernel functions defined on the training samples,

$$\hat{p}(\boldsymbol{x}_*) = \frac{1}{n}\sum_{i=1}^{n} k(\boldsymbol{x}_*, \boldsymbol{x}_i) = \frac{1}{n}\boldsymbol{k}_* \mathbb{1}_n, \tag{18}$$

where $\boldsymbol{k}_*$ is the vector of kernel evaluations between the point of interest $\boldsymbol{x}_*$, and all training samples (see section 3.1). KDE kernel functions have to be non-negative and integrate to one to ensure that $\hat{p}$ is a valid pdf. When a point-dependent weighting, $\beta_i$, is employed, then the above expression can be modified as $\hat{p}(\boldsymbol{x}_*) = \sum_{i=1}^{n} \beta_i k(\boldsymbol{x}_*, \boldsymbol{x}_i)$, where the $\beta_i$ have to be positive and sum to one, i.e. $\beta_i \geq 0$ and $\sum_{i=1}^{n} \beta_i = 1$. In [51] a solution to find a suitable $\boldsymbol{\beta}$ vector based on kernel principal components analysis was proposed. If the decomposition of the un-centered kernel matrix follows the form $\mathbf{K} = \mathbf{EDE}^{\top}$, where $\mathbf{E}$ is orthonormal and $\mathbf{D}$ is a diagonal matrix, then the kernel-based density estimation can be expressed as

$$\hat{p}(\boldsymbol{x}_*) = \boldsymbol{k}_* \mathbf{E}_r \mathbf{E}_r^{\top} \mathbb{1}_n, \tag{19}$$

where $\mathbf{E}_r$ is the reduced version of $\mathbf{E}$ by keeping $r < n$ top eigenvectors. If we keep all the dimensions, i.e. $r = n$ the solution reduces to (18). By reducing the number of components we restrict the capacity of the density estimator and hence obtain a smoother approximation of the pdf as $r$ reduces.

The retained kernel components should be selected by keeping the dimensions that maximize a sensible pdf characteristic, e.g. the variance. However, other criteria can be used to select the retained components. For instance, the kernel entropy component analysis (KECA) method uses the *information potential* as criterion to select the components from the eigenvector decomposition [8]. In this case, the decomposition method is already optimized to maximize the variance, therefore the solution will be sub-optimal. A more accurate way of finding a decomposition was presented in [55] where the features are directly optimized to maximize the amount of retained information. This method was named optimized KECA (OKECA), and showed excellent performance using very few extracted components.

The relevant aspect for this paper is that, by doing $\boldsymbol{\alpha} = \mathbf{E}_r \mathbf{E}_r^{\top} \mathbb{1}_n$, Eqs (18) and (19) can be cast in the general framework of kernel methods we proposed in Eq (4). Through this equality the derivatives and the second derivatives (and therefore the Hessian) can be obtained in a straightforward manner using Eqs (5) and (6). This information can be used for different problems, such as computing the Fisher's information matrix, optimizing vector quantization systems, or the example in the following section where we use them to find the points that belong to the principal curve of the distribution.

## 5.2 Derivatives and principal curves

This example illustrates the use of kernel derivatives in the KDE framework. In particular, we use the gradient and the Hessian of the pdf, to find points that belong to the *principal curve* along the data manifold [56]. A principal curve is defined as the curve that passes through the middle of the data. How to find this curve in practice is an important problem since multiple data description methods are based on drawing principal curves [30, 57–60]. In [30], they characterize the principal curve as the set of points that belong to the ridge of the density function. These points can be determined by using the gradient and the Hessian of the pdf: a point $\boldsymbol{x}_*$ is an element of the $d$-dimensional principal curve iff the inner product of the gradient, $\nabla \hat{p}(\boldsymbol{x}_*)$, and at least $r$ eigenvectors of the Hessian, $\mathbf{H}(\boldsymbol{x}_*)$, is zero:

$$\nabla \hat{p}(\boldsymbol{x}_*)^{\top} \mathbf{E}_r(\boldsymbol{x}_*) = \mathbf{0}, \tag{20}$$

where $\mathbf{E}_r(\boldsymbol{x}_*)$ are the top $r$ eigenvectors of the matrix $\mathbf{H}(\boldsymbol{x}_*)$. Note that applying this definition using our framework is straightforward as we can use the KDE to describe the probability density function, and Eqs (5) and (6), as well as formulas in Table 1, to find the gradient and the Hessian of the defined pdf with respect to the points. See Table 3 for a comparison to other kernel methods derivatives.

In [Fig 4](), we show an illustrative example of this application in three different toy datasets. The pdf can be obtained from the data points by using the OKECA method and the derivative lines describe the direction to which the density changes the most. The last row shows the points of the dataset with smaller dot products between the gradient and the last eigenvector of the Hessian, see [Eq (20)](). Note that these points belong to the ridge of the distribution, and thus to the principal curve.

## 6 Kernel dependence estimation

### 6.1 Dependence estimation with kernel methods

Measuring dependencies and nonlinear associations between random variables is an active field of research. The kernel-based dependence estimation defines a covariance and cross-covariance operators in RKHS, and the subsequent statistics from these operators allows one to measure dependence between functions therein.

Let us consider two spaces $\mathcal{X} \subseteq \mathbb{R}^{d_x}$ and $\mathcal{Y} \subseteq \mathbb{R}^{d_y}$, which we jointly sample observation pairs $(\boldsymbol{x}, \boldsymbol{y})$ from distribution $\mathbb{P}_{xy}$. The covariance matrix is $\mathcal{C}_{xy} = \mathbb{E}_{xy}(\boldsymbol{x}\boldsymbol{y}^\top) - \mathbb{E}_{x}(\boldsymbol{x})\mathbb{E}_{y}(\boldsymbol{y}^\top)$, where $\mathbb{E}_{xy}$ is the expectation with respect to $\mathbb{P}_{xy}$, and $\mathbb{E}_{x}$. A statistic that summarizes the content of the covariance matrix is its Hilbert-Schmidt norm. This quantity is zero if and only if there exists no second order dependence between $\boldsymbol{x}$ and $\boldsymbol{y}$.

The nonlinear extension of the notion of covariance was proposed in [13] to account for higher order statistics. Essentially, let us define a (possibly non-linear) mapping $\phi : \mathcal{X} \to \mathcal{F}$ such that the inner product between features is given by a PSD kernel function $k(\boldsymbol{x}, \boldsymbol{x}')$. The feature space $\mathcal{F}$ has the structure of a RKHS. Similarly, we define $\psi : \mathcal{Y} \to \mathcal{G}$ with associated kernel function $l(\boldsymbol{y}, \boldsymbol{y}')$. Then, it is possible to define a cross-covariance operator between these feature maps, and to compute the squared norm of the cross-covariance operator, $\| \mathcal{C}_{xy} \|_{\text{HS}}^2$, which is called the Hilbert-Schmidt Independence Criterion (HSIC) and can be expressed in terms of kernels [61, 62]. Given a sample dataset $\mathcal{D} = \{(\boldsymbol{x}_1, \boldsymbol{y}_1), \ldots, (\boldsymbol{x}_n, \boldsymbol{y}_n)\}$ of size $n$ drawn from $\mathbb{P}_{xy}$, an empirical estimator of HSIC is [13]:

$$\text{HSIC}(\mathcal{F}, \mathcal{G}, \mathbb{P}_{xy}) = \frac{1}{n^2}\text{Tr}(\mathbf{KHLH}) = \frac{1}{n^2}\text{Tr}(\mathbf{HKHL}), \tag{21}$$

where $\text{Tr}(\cdot)$ is the trace operation, $\mathbf{K}$, $\mathbf{L}$ are the kernel matrices for the input random variables $\boldsymbol{x}$ and $\mathbf{y}$ (i.e. $[\mathbf{K}]_{ij} = k(\boldsymbol{x}_i, \boldsymbol{x}_j)$), respectively, and $\mathbf{H} = \mathbf{I} - \frac{1}{n}\mathbb{1}\mathbb{1}^\top$ centers the data in the feature spaces $\mathcal{F}$ and $\mathcal{G}$, respectively. HSIC has demonstrated its capability to detect dependence between random variables but, as for any kernel method, the learned relations are hidden behind the kernel feature mapping. To address this issue, we consider the derivatives of HSIC.

### 6.2 Derivatives of HSIC

HSIC empirical estimate is parameterized as a function of two random variables, so the function derivatives given in section 2 are not directly applicable. Since HSIC is a symmetric measure, the solution for the derivative of HSIC wrt $x_i^j$ will have the same form as the derivative wrt $y_i^j$. For convenience, we can group all terms that do not explicitly depend of $\mathbf{X}$ as $\mathbf{A} = \mathbf{HLH}$, which allows us expressing (21) simply as:

$$\text{HSIC} := \frac{1}{n^2}\text{Tr}(\mathbf{KA}) = \frac{1}{n^2}\sum_{i=1}^{n}\sum_{j=1}^{n}[\mathbf{A}]_{ij}k(\boldsymbol{x}_i, \boldsymbol{x}_j). \tag{22}$$

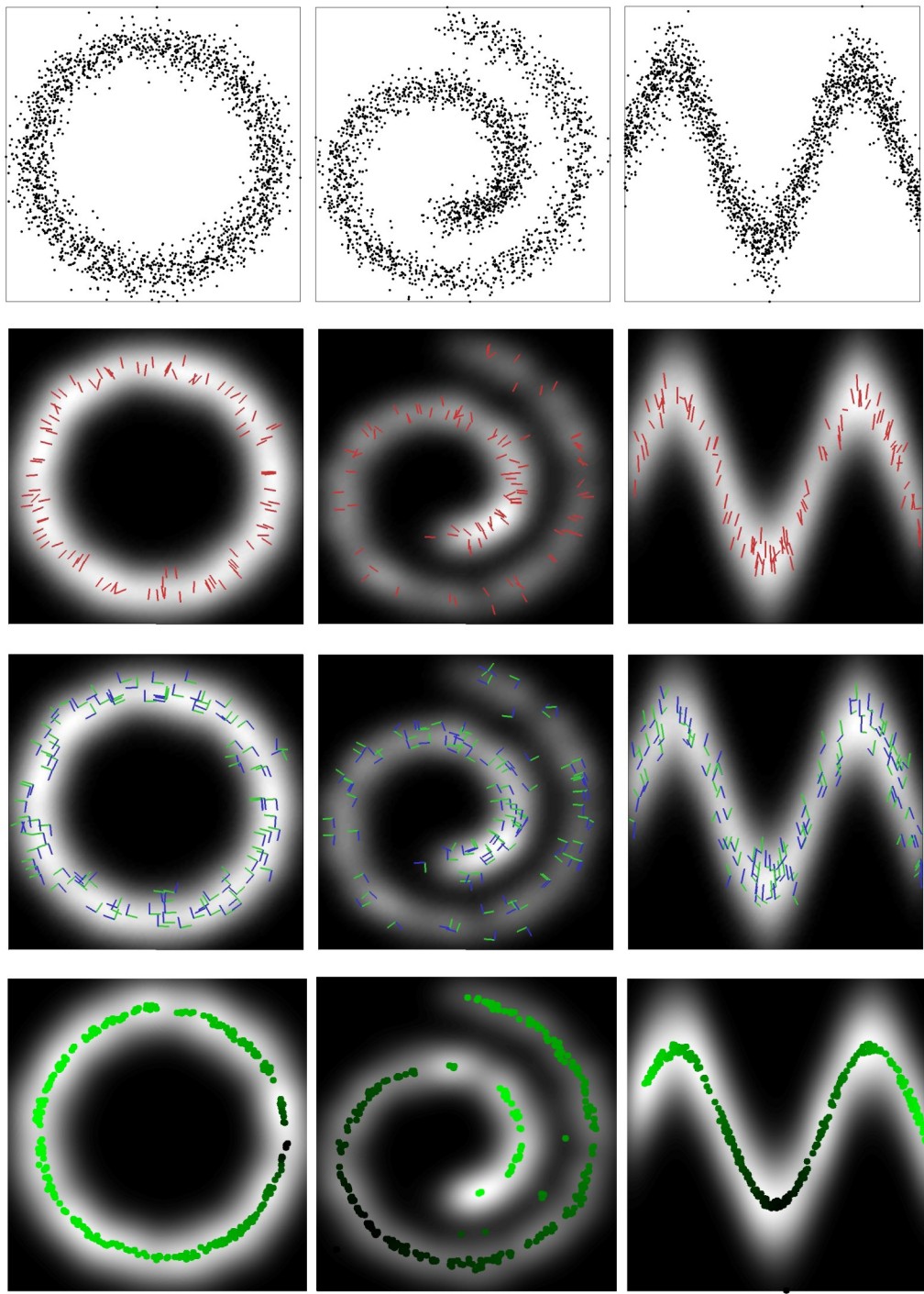

**Fig 4. First row**: Original data points. Second, third and fourth row: probability density in gray scale (brighter means denser). **Second row**: derivative direction of the pdf for some data points is represented using red lines. **Third row**: Hessian eigenvectors for some points represented with blue lines (first eigenvector) and green lines (second eigenvector). **Fourth row**: points on the ridge computed using the formula proposed in [30], different brightness of green has been computed using the Dijkstra distance over the curve dots (see text for details).

Note that the core of the solution is the same as in the previous sections; a weighted combination of kernel similarities. However, now we need to derive both arguments of the kernel function $k$ with respect to entry $x_i^j$ that appears twice. By taking derivatives with regards to a particular dimension $q$ of sample $\boldsymbol{x}_i$, i.e. $x_i^q$, and noting that the derivative of a kernel function is a symmetric operation, i.e. $\frac{\partial k(\boldsymbol{x}_i, \boldsymbol{x}_j)}{\partial x_i^q} = \frac{\partial k(\boldsymbol{x}_j, \boldsymbol{x}_i)}{\partial x_i^q}$, one obtains

$$\frac{\partial \mathrm{HSIC}}{\partial x_i^q} = \frac{2}{n^2} \sum_{j=1}^{n} [\mathbf{A}]_{ij} \frac{\partial k(\boldsymbol{x}_i, \boldsymbol{x}_j)}{\partial x_i^q} = \frac{2}{n^2} \mathbf{A}_i \partial_q \boldsymbol{k}(\mathbf{x}_i), \tag{23}$$

where $\mathbf{A}_i$ is the $i$-th row of the matrix $\mathbf{A}$. For the the RBF kernel we obtain [31]:

$$\frac{\partial \mathrm{HSIC}}{\partial x_i^q} = -\frac{2}{\sigma^2 n^2} \mathrm{Tr}(\mathbf{HLH}(\mathbf{K} \circ \mathbf{M}^q)), \tag{24}$$

where entries of matrix $\boldsymbol{M^q}$ are $[\boldsymbol{M^q}]_{ij} = x_i^q - x_j^q$ $(1 \leq j \leq n)$, and zeros otherwise, and where the symbol $\circ$ is the Hadamard product between matrices.

Recently [63] extended the notion of leverage scores for the ridge regression problem. Leverage is a measure of how points with low density neighbours are enforcing the model for passing through them. By definition, the leverage (of a regressor) is the sensitivity of the predictive function w.r.t. the outputs. There is no definition of leverage in the case of HSIC as it is not a regression model but a dependence measure. However, HSIC could be interpreted in a similar way by fixing one of the variables and taking the derivative w.r.t. the other. By this interpretation, one can think of the HSIC sensitivity as a measure of how individual points are affecting the dependence measurement, i.e. how sensitive HSIC is to the perturbations for each particular point. This interpretation allows us to link the concepts of leverage and sensitivity in kernel dependence measures.

In this case, the derivatives of HSIC report information about the directions that impact the dependence estimate the most. This allows one to evaluate the measure as a *vector field* representation of two components. As in the previous kernel methods analyzed, the derivatives here are also analytic, just involving simple matrix multiplications and a trace operation. See Table 3 for a comparison to other kernel methods derivatives.

## 6.3 Visualizing kernel dependence measures

HSIC derivatives give information about the contribution of each point and feature to the dependence estimate. Fig 5 shows the directional derivative maps for three different bi-dimensional problems of variable association. We show the different components of the (sign-valued) vector field as well as its magnitude. In all problems, arrows indicate the strength of distortion to be applied to points (either in directions $x$, $y$, or jointly) such that the dependence is maximized. For the first example (top row), the map pushes the points into the 1-1 line and tries to collapse data into 2 different clusters along this line. In the second example (middle row), the distribution is a noisy ring: here the sensitivity map tries to collapse the data into clusters in order to maximize the dependence between the variables. In the last third experiment (bottom row), both variables are almost independent and the sensitivity map points towards some regions in the space where the dependence is maximized. In all cases, the $S_x$ and $S_y$ are orthogonal in direction and form a vector field whose intensity can be summarized in its norm $|S|$ (columns in the figure).

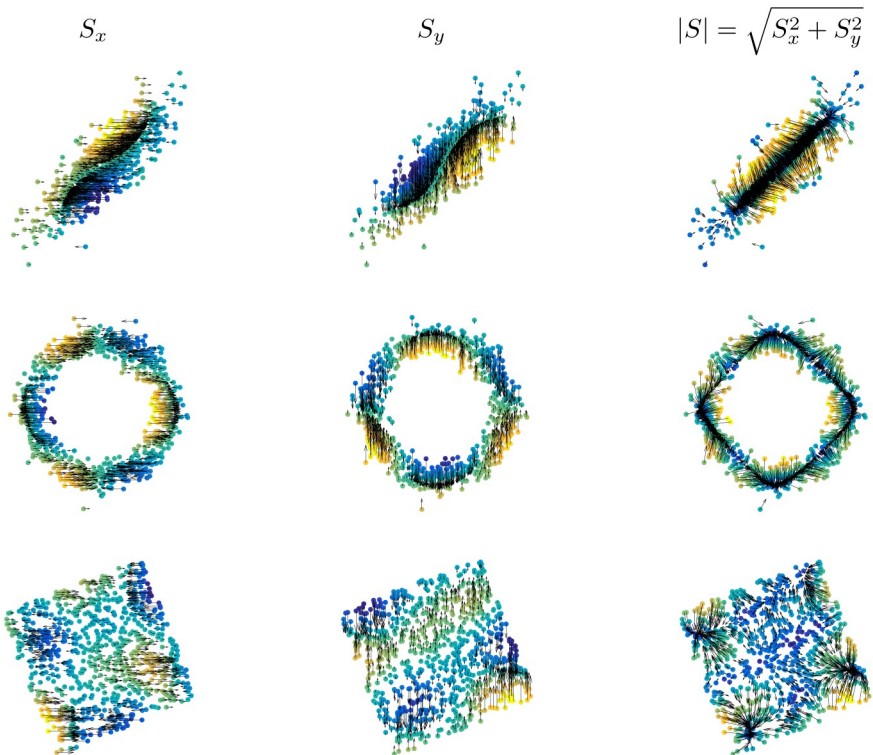

**Fig 5. Visualizing the derivatives and the modulus of the directional derivative for HSIC in three toy examples.**

## 6.4 Unfolding and independization

We have seen that the derivatives of the HSIC function can be useful to learn about the data distribution and the variable associations. The derivatives of HSIC give information about the directions most affecting the dependence or independence measure.

Fig 6 shows an example of how the derivatives of the HSIC can be used to modify the data and achieve either maximum dependence or maximum independence. We embedded the derivatives in a simple gradient descent scheme, in which we move samples iteratively to maximize or minimize data dependence. Departing from a sinusoid, one can attain dependent or independent domains.

Note that HSIC can be understood as a maximum mean discrepancy (MMD) [64] between the joint probability measure of the involved variables and the product of their marginals, and MMD derivatives are very similar to those of HSIC provided here. The explicit use of the kernel derivatives would allow us to use gradient-descent approaches in methods that take advantage of HSIC or MMD, such as in algorithms for domain adaptation and generative modeling.

## 7 Analysis of spatio-temporal earth data

Kernel methods are widely applied in the Earth system sciences [5], where they have proven to be effective when dealing with low numbers of (potentially high dimensional) training samples. Data of this kind are characteristic for hyperspectral data, multidimensional sensor information, and different noise sources in the data. The most common applications in Earth system sciences are anomaly and target detection [65], the estimation of biogeochemical or biophysical parameters [66–68], dimensionality reduction [15, 69, 70], and the estimation of data interdependence [31]. However, so far multivariate spatio-temporal data problems have

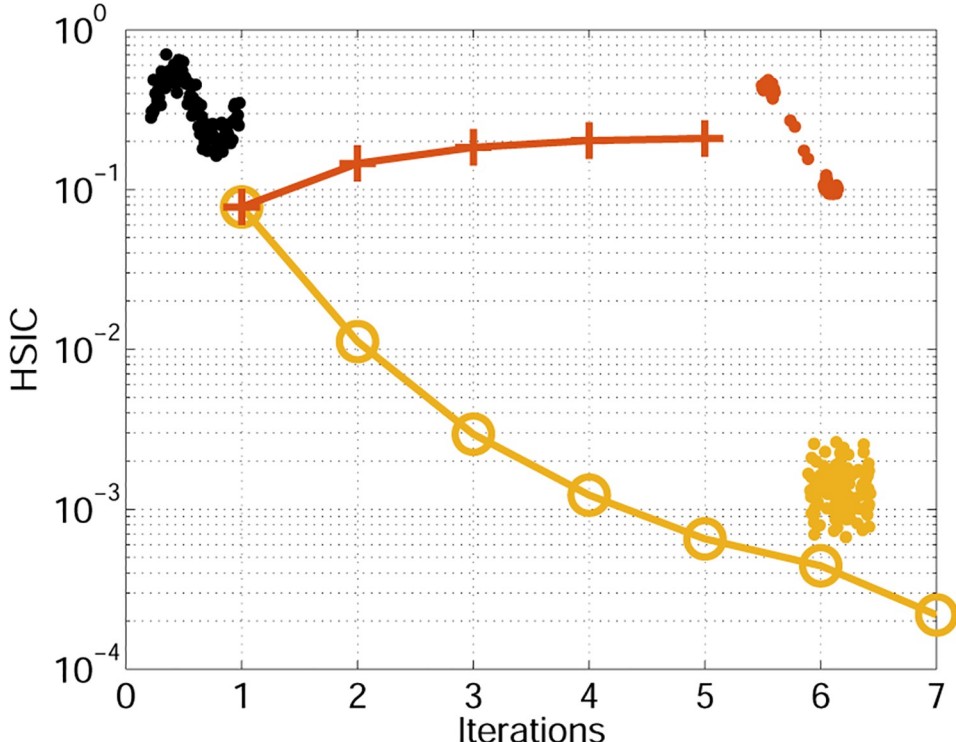

**Fig 6. Modification of the input samples to maximize of minimize HSIC dependence between their dimensions (see text for details).**

received comparable little attention [71, 72], and in particular regarding the use of the derivatives of kernel methods [25, 26]. This is surprising, given the high-dimensional nature of most spatio-temporal dynamics in most sub-domains of the Earth system, e.g. land-surface dynamics, land-atmosphere interactions, ocean dynamics, etc. [73]. Hence, this section explores the added value of kernel derivatives for analyzing multivariate spatio-temporal Earth system data. We showcase applications considering the four studied problems of classification, regression, density estimation and dependence estimation. Please see github.com/IPL-UV/sakame for a working implementation of the algorithms as well as the subsequent ESDC experiments.

## 7.1 Spatio-temporal earth data

Today, data-driven research into Earth system dynamics has gained momentum and complements global modelling efforts. Much of Earth data is generated by a wide range of satellite sensors, upscaled products from *in-situ* observations, and model simulations with constantly improving spatial and temporal resolutions. The question is whether using kernel derivatives may help in (1) choosing the appropriate space and time scales to analyze phenomena, (2) visualize the most informative areas of interest, and (3) detect anomalies in spatio-temporal Earth data. We will work with products contained in the Earth System Data Lab (ESDL) [73]. The analysis-ready data-cube contains and harmonizes more than 40 variables relevant to monitor key processes of the terrestrial land-surface and atmosphere. The data streams contained in the ESDL are grouped in three data streams: land surface, atmospheric forcings and socio-economic data. Here we focus on three land-surface variables which exhibit nonlinear

relations in space and time. The following three variables; the gross primary productivity (GPP), root-zone soil moisture (SM), and land surface temperature (LST); are outlined below:

- GPP is the rate of fixation of carbon dioxide through the photosynthesis and one of the largest single flux in the global carbon cycle. However, the process is sensitive to climate variability. For instance, it has been shown that regional extreme events like droughts, heatwaves, and other types of disturbances may even influence the inter-annual variability of the globally integrated GPP [74]. Hence, it is key to understand the spatial and temporal dynamics of GPP at regional and global scales. Here, we consider the GPP FLUXCOM (http://www.fluxcom.org/) product, computed as described in [75, 76].

- **SM** plays a fundamental role for the environment and climate system, as it influences hydrological and agricultural processes, runoff generation and drought development processes, and land-atmospheric feedbacks [77] There are two products of soil moisture in our experiments. Standard SM products carry information limited to a few centimeters below the surface (±5 cm), and do not allow access to the whole zone from where water can be absorbed by roots. This is why we used root-zone soil moisture (RSM) [78–80] in the dependence estimation problem instead, a product from GLEAM that is a more sensitive variable to monitor water stress and droughts in vegetation.

- **LST** is an essential variable within the Earth climate system as it influences processes such as the exchange of energy and water between the land surface and atmosphere, and influences the rate and timing of plant growth. The LST product contained in the ESDL is the result of an ESA project called GlobTemperature, that developed a merged LST data set from thermal infrared (geostationary and polar orbiters) and passive microwave satellite data to provide best possible coverage.

The data is organized in 4-dimensional data cube $\mathbf{x}(u, v, t, k)$ involving (latitude, longitude) spatial coordinates $(u, v)$, time sampling $t$, and the variable $k$. The data in ESDL contains a spatial resolution (high 0.083˚ resolution and coarser grid aggregation at 0.25˚) and a temporal resolution of 8 days spanning the years 2001-2011. In our experiments, we focus on the lower resolution products, during 2008-2010, and over Europe only. In the year 2010, a severe combination of spring and summer drought combined with a summer heat stress event affected large parts of Russia which can be observed in the three variables under study here [81], and we expect that also their interrelations must be affected. We use this well known event to provide a proof of concept for our suggestion approaches to interpret regressions, principal curves, and dependence estimation.

## 7.2 Sensitivity analysis in GP modelling

Studying time-varying processes with GPs is customary. Designing a GP becomes more complicated when dealing with spatio-temporal datasets. This can be cumbersome when the final goal is to understand and visualize spatial dependencies as well as to study the relevance of the features and the samples. Sensitivity analysis can be useful for either scenario. In this experiment, we study the impact of features in the GP modeling of the GPP and LST variables during 2010. To do so, we developed GP regression models trained to predict a pixel from their neighbourhood pixels. This is similar to geographically weighted regression [82] which can be used to model the local spatial relationships between these features and the outputs. From this framework, we can get sensitivity values for each of the contributing dimensions. We further split the data into subsets of spatial 'minicubes' which ranged in size from $2 \times 2$ until size $7 \times 7$. We use a GP model on a training subset of minicubes whereby the neighbours were used as

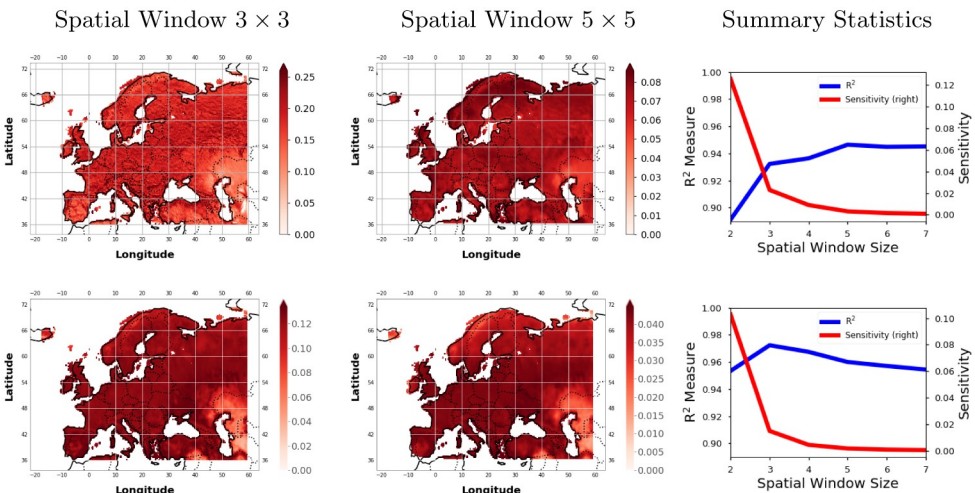

**Fig 7. Visualizing the spatial maps for the senstivity of the Gaussian process (GP) regression model under different spatial sampling sizes for the Gross Primary Productivity (GPP) [top] and land surface temperature (LST) [bottom] for the summer of 2010 (Jun-Jul-Aug).** The rightmost column shows the summary $R^2$ and Sensitivity for each spatial window size for the GP model.

input dimensions to predict the center pixel for both GPP and LST. For metrics, we used the $R^2$-value to measure the goodness of fit between our model and the real data.

Fig 7 show how the sensitivity changes according to the mean prediction of the GPP and LST for two neighbourhood spatial window sizes ($3 \times 3$ and $5 \times 5$). It also shows the spatial sensitivity maps for both settings and the $R^2$-value and the average sensitivity for each GP model. What's reassuring is that we see consistently low sensitivity values in areas (e.g. near the Black and Caspian Sea) for the GPP and LST regardless of the spatial window size as these are typically areas of low GPP and SM. For GPP, we see that sensitivities tend to become smoother as the neighbourhood size increases. These particular maps for GPP reach an $R^2$ value of 0.93 and 0.95 for each respective window size. Unlike the small differences in goodness of fit (+2% in $R^2$), the sensitivity curves show a wider variation and suggest that bigger windows are more appropriate to capture smoother areas; this is expected. Although we get a better model with a higher spatial window size, the sensitivity of neighbouring points become more dispersed over larger areas over Europe instead of just staying within small clusters. A similar pattern of dispersion of the sensitive points is observed for the LST maps w.r.t. the spatial window size. For LST, we notice that there is not a large difference in the $R^2$ as we increase the spatial window size. The most sensitive regions mostly stay the same but there is a small shift from the northern regions of Europe from more sensitive to less sensitive. So it's clear that the number of spatial-pixels used as input features would be different depending upon the input variable, e.g. one can use a higher neighbourhood size for LST because we get the same $R^2$ and similar sensitivity maps whereas the GPP could have a lower window size to ensure that we capture the local variability.

## 7.3 Classification of drought regions

Support vector machines (SVMs) is a very common classification method widely used in numerous applications in the field of machine learning in general and remote sensing in particular [5]. The derivatives of the SVM function, however, have not been used before to understand the model, nor linked to the concept of margin. The derivatives of SVMs can be broken

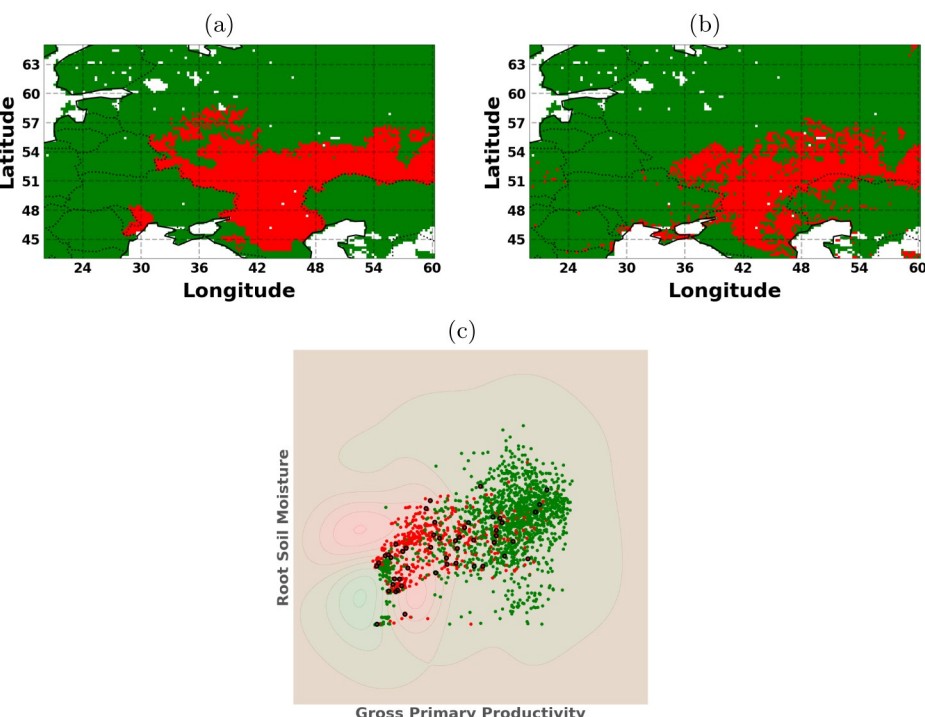

**Fig 8. Visualizing the (a) labels, (b) predictions (b), and (c) the 2D representation space for the predictions.** This is the classification problem of drought (red) versus no-drought (green) with the support vectors (black) for the SVM formulation (section 4).

into two components via the product rule (section 4): 1) the derivative of the mask function and 2) the derivative of the kernel function. Typically one would use a sgn function as the mask but we used the *tanh* function to allow us to observe how the boundary or margin behaves w.r.t. the inputs.

In this experiment, we chose to study the relationship between gross primary productivity and root soil moisture during the year 2010 affected by a severe heatwave. There was a severe drought that occurred over this region for the entire summer of 2010 (June, July and August) [81]. Drought classification is an unsupervised problem and so there is a lot of debate about how to detect droughts within different scientific communities. We use a pre-defined drought mask of the countries affected by the 2010 heatwave found from the EM-DAT database [83] which reports all drought events which follow at least one of the criteria: 10 or more people dead, 100 or more people affected, declared state of emergency, or a call for international assistance. The region where the droughts are reported is just over the region of Eastern Europe, as shown in the binary classification maps in Fig 8. We chose this pre-defined drought area to simplify the problem which would allow us to see if we can indeed classify a drought region spatially and then look at the derivatives. We did a simple binary classification problem over the spatial coordinates using the two input variables (GPP and RM). We sampled only from the month of July at different time intervals within the month to make the samples more varied as the GPP and RM can still fluctuate within a monthly span. This is an unbalanced dataset as there are more non-drought regions than drought regions in the spatial subsample. While there are numerous advanced methods to deal with imbalanced datasets, we only used the standard SVM as that complexity is out of the scope for this experiment. The ESDC is very dense so we used 500 randomly selected points for the drought region and 1,000 randomly

**Table 4. This table summarizes classification results for the drought and non-drought regions over Eastern Europe using the SVM (Support Vector Machines, section 4) formulation.**

| Class Label | Precision | Recall | F1-Score | Support |
|---|---|---|---|---|
| Non-Drought Regions | 0.90 | 0.91 | 0.91 | 28590 |
| Drought Regions | 0.69 | 0.67 | 0.68 | 8268 |
| **Accuracy** | | | 0.86 | 36858 |

selected points for the non-drought regions. The remaining points ($\sim$ 3800) were considered for calculating test statistics, while the visualizations include all of the points for the dataset. We applied a standard cross-validated SVM classification algorithm with an RBF kernel function. For metrics, we used the standard precision, recall, F1-score and Support for the predictions of drought over non-drought. Table 4 shows the classification results compared to the labels of the trained SVM algorithm and Fig 8 shows the classification maps.

Fig 9 shows the sensitivity spatial maps as well as the 2D latent space for the outputs of the SVM classification model. We show the full derivative and the mask and kernel product components. The mask derivative has high sensitivity values for almost all regions where the decision function is unsure about the classification region. We see that the highest yellow regions are near the Caspian Sea which is also the area where there is a lot of overlap

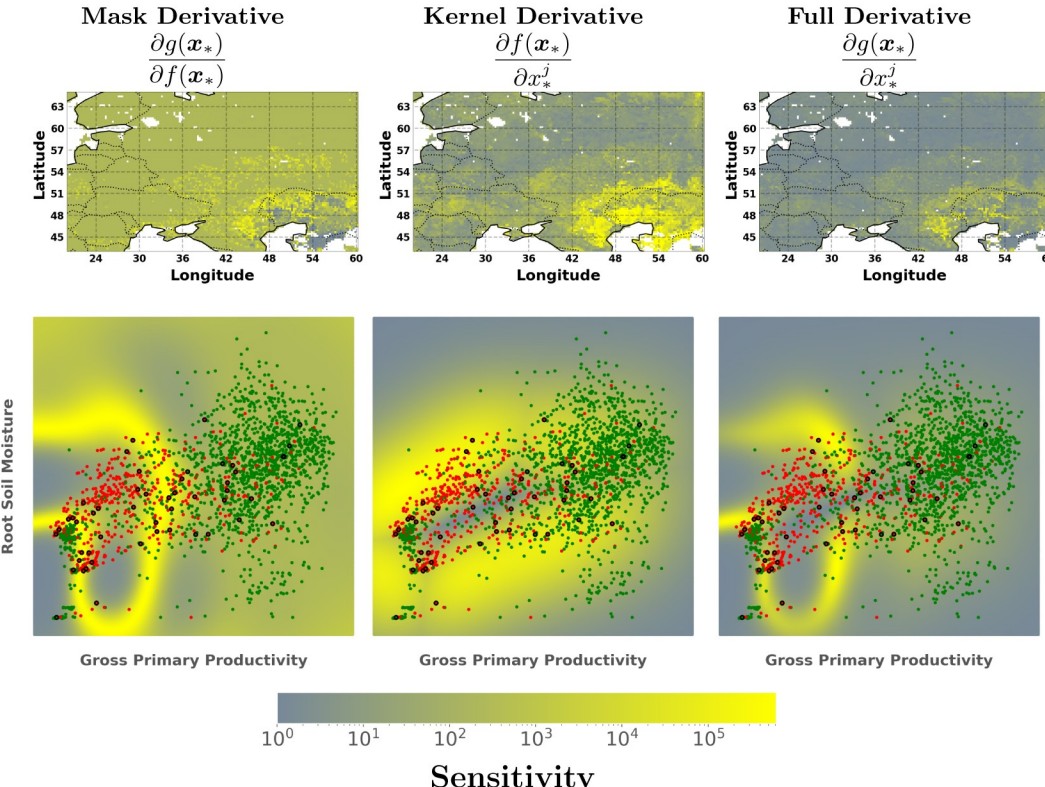

**Fig 9. Visualizing the scatter plot of the drought (red) versus no-drought (green) and the support vectors (black) using the SVM section 4 classification algorithm.** We also display the sensitivity of the full derivative and its components: the mask function (tanh) and the kernel function ($\partial \mathbf{k} \cdot \alpha y$) based on the predictive mean of the SVM classification results.

between the classes. Recall that the mask used is based on the reported regions and not the actual GPP or SM values. So naturally the SVM algorithm is probably picking up the inconsistencies with the data given. Nevertheless, the kernel derivative indicates where there are regions of little data and in regions where there is significant overlap. Ultimately, the combination of the products represents a good balance between lack of data and the width of the margin between the two classes.

## 7.4 Principal curves of the ESDC

In this experiment we analyze GPP spatial-temporal patterns for different seasons for the year 2010 using the *principal curve* (PC) framework in Section 4. Each sample consists of a vector with the variable value for a particular location and all the time dimensions in the season: *(05-Jan to 05-May)*, *(21-May to 08-Aug)*, and *(17-Aug to 31-Dec)* For each season we have around 28, 000 samples of size $1 \times T$. Fig 10 shows the results. For each data set we plot the mean GPP value of the season in each point. The location of the points that belong to the PC are plotted in green using the Dijkstra distance inside the curve (as in the toy examples in Fig 4). The points belonging to the PC can be interpreted as the landmarks of the whole dataset, similar to a centroid of a cluster. But in this example, they refer to the points on the probability ridge of the data manifold (i.e. similar to the points closer to the first eigenvector in PCA). These points could be used for multiple purposes, e.g. as a summary to analyze the behaviour of the whole manifold or used for a temporal analysis of their evolution. One one hand, the location of the points is quite independent of the mean values, so they give different, alternative information. On the other hand, the location depends on the time of the year represented.

Most of the GPP 'representative' points are scattered around the manifold which depends on the season. For instance during the colder season (Jan-May), the dots are concentrated in the middle and low latitudes. During this period, the dots in northern Germany have a similar temperature and GPP than in the North-West part of Europe. Therefore, there is no need to add extra landmarks in these regions. Points in Morocco represent the warmer part of the manifold and Balcans area and Turkey represent the central part of the manifold. During the warmest period (May-Aug) the distribution of the dots follow an opposite direction, Southern regions are weighted less while Northern regions have more representation. In the case of mild temperatures (Aug-Dec), more landmarks in different regions are needed.

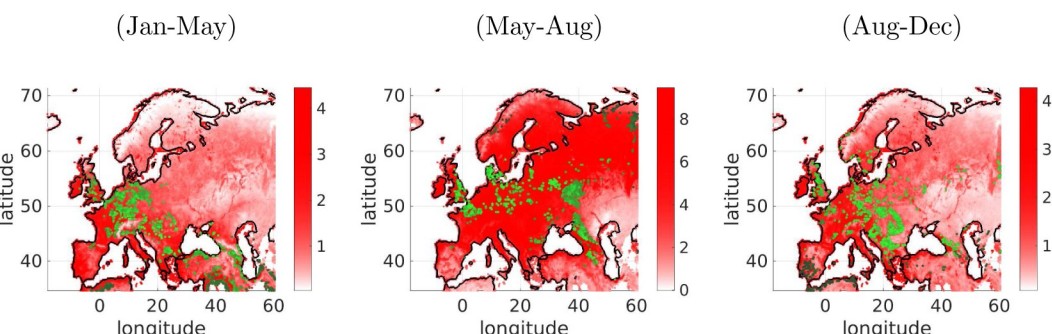

**Fig 10. Principal curves on the ESDC.** Each figure represents the results for GPP at different time periods during the 2010. In each image the mean value of the variable for each location is shown in colormap (minimum blue, maximum red), and the points that belong to the principal curves are represented in green. Different brightness of green has been computed using the Dijkstra distance over the curve dots.

### 7.5 Sensitivity analysis of kernel dependence measures

HSIC is a dependence measure which can show differences in the higher-order relations between random variables. The derivatives of HSIC w.r.t. the input features are related to the change of the dependence measure which summarizes the relevance of the input features in the dependence. Therefore, these derivative maps can be related to the *sensitivity* of the inputs.

In this experiment, we chose to study the relation between GPP and RM for Europe and Russia during the years 2008, 2009 and 2010. We apply the HSIC with a linear kernel and compute the sensitivity maps, which is an estimation of how much the dependency criterion changes. We take spatial segments of GPP and RM at each time stamp, $T$ and compute the HSIC value for each $T$ independently for Russia and Europe. We also computed the derivative of HSIC for the same $T$ time stamps independently for Russian and Europe. We computed the modulus to summarize the impact of each dimension to act as a proxy for the total average sensitivity. The final step involved computing the expectation between the modulus of the derivative of HSIC between Russian and Europe. Europe acts as a proxy stable environment and Russia is the one we would like to compare to. We estimated the expected value for three time periods (before: 05-Jan, 20-May; during: 28-May, 01-Sep; after: 09-Sep, 30-Dec) for each year individually. Then we compared each of the values to see how the expectation changes between Europe and Russia for each period across the years. The expected value of the HSIC derivatives summarize the change of association between variables differently than the HSIC measure itself.

The experiment focuses on studying the coupling/association between RM and GPP during the Russian drought in 2010. The HSIC algorithm captures an increased difference in dependencies of GPP and RM for Russia relative to Europe in 2010 if we compare this relationship to the years 2008 and 2009, see Fig 11a. However, HSIC only captures instantaneous instances of dependencies and not how fast these changes occur. The derivatives of HSIC (Fig 11b) allow us to quantify and capture when these changes actually occur. The gradients of HSIC do not show obvious differences in magnitude or shape across years between Russia and Europe. By taking the expected value of specific time periods of interest (before-during-after drought), we can highlight the contrast in the dependency trends between different periods with respect to their previous years, both in terms of HSIC and HSIC derivatives. We observe in Fig 11c, a change the mean value of the difference in the derivative of HSIC in Fig 11d which reveals a noticeable change in the trend for the springtime and summertime of 2010 compared to 2008 and 2009.

## 8 Conclusions

The use of Kernel methods is very popular in pattern analysis and machine learning and have been widely adopted because of their performance in many applications. However, they are still considered black-box models as the feature map is not directly accessible and predictions are difficult to interpret. In this note, we took a modest step back to understand different kernel methods by exploiting partial derivatives of the learned function with respect to the inputs.

To recap, we have provided intuitive explanations for derivatives of each kernel method through illustrative toy examples, and also highlighted the links between each of the formulations with concise expressions to showcase the similarities. We show that 1) the derivatives of kernel regression models (such as GPs) allows one to do sensitivity analysis to find relevant input features, 2) the derivatives of kernel classification models (such as SVMs) also allows one to do sensitivity analysis and visualize the margin, 3) the derivatives of kernel density estimators (KDE) allows one to describe the ridge of the estimated multivariate densities, 4) the derivatives of kernel dependence measures (such as HSIC) allows one to

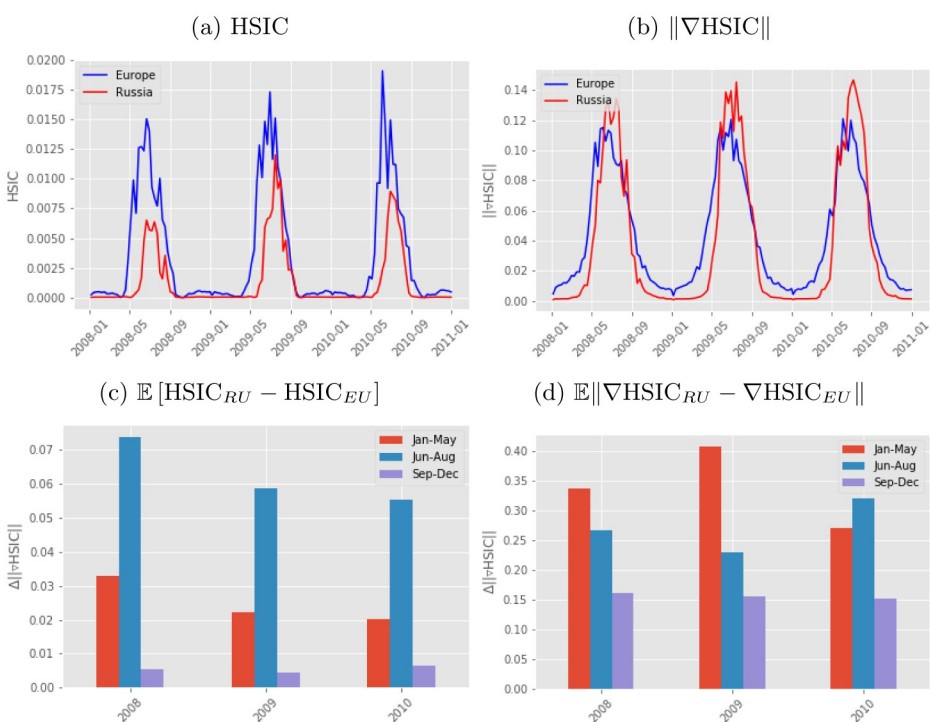

**Fig 11. Each figure represents different summaries of how HSIC can be used to capture the differences in dependencies between Europe and Russia for GPP and RSM.** (a) shows the HSIC value for Europe and Russia at each time stamp, (b) shows the derivative of HSIC for Europe and Russia at each time stamp, and the mean value for the difference in the (c) HSIC between Europe and Russia for different periods (Jan-May, Jun-Aug, and Sept-Dec), and (d) in the derivative of HSIC for the same periods.

visualize the magnitude and change of direction in the dependencies between two multivariate variables. We have also given proof-of-concept examples of how they can be used in challenging applications with spatial-temporal Earth datasets. In particular, 1) we show that we can express the spatial-temporal relationships as inputs to regression algorithms and evaluate their relevance for prediction of essential climate variables, 2) we show that we can assess the margin for classification models in drought detection, as a way to identify the most sensitive points/regions for detection, 3) we show that the ridges can be used as indicators of potential regions of interest due to their location in the PDF, which could be related to anomalies, and 4) we show that we can detect changes in dependence between two events during an extreme heatwave event.

## A Higher order derivatives of kernel functions

It can be shown that the $m$-th derivative of some kernel functions can be computed recursively using Faà di Bruno's identity [40] for the multivariate case:

$$\frac{\partial^{(m)}}{\partial x^{j^{(m)}}}f(g(\boldsymbol{x})) = \sum \frac{m!}{t_1!\,1!^{t_1}\,t_2!\,2!^{t_2}\cdots t_m!\,m!^{t_m}} \cdot \frac{\partial^{(t_1+\cdots+t_m)}f}{\partial g(\boldsymbol{x})} \cdot \prod_{i=1}^{n}\left(\frac{\partial g^{(i)}}{\partial x^{j^{(i)}}}\right)^{m_j},$$

where the sum is over all $m$-tuples $(t_1,\ldots,t_m) \in \mathbb{N}^m$ and $\sum_{j=1}^{m} j\,t_j = m$. It is also useful the

expression for mixed derivatives:

$$\frac{\partial^{(m)}}{\partial x^1 \cdots \partial x^m} f(g(\boldsymbol{x})) = \sum_{\pi \in \Pi} \frac{\partial^{|\pi|} f}{\partial g(\boldsymbol{x})} \cdot \prod_{B \in \pi} \frac{\partial g^{|B|}}{\prod_{j \in B} \partial x^{|B|}},$$

where $\Pi$ is the ensemble all the partitions sets in $1 \ldots m$, $\pi$ is a particular partition set, $B \in \pi$ runs over the blocks of the partition set $\pi$, and $|\pi|$ is the cardinality of $\pi$.

For the RBF kernel we can identify $f = \exp(\cdot)$ and $g = -\gamma\|\boldsymbol{x} - \boldsymbol{y}\|^2$. The derivatives for the $f(g(\boldsymbol{x}))$ are always the same $\partial^m f / \partial g(\boldsymbol{x})^m = f(g(\boldsymbol{x})) = \exp(g(\boldsymbol{x}))$, and the derivatives for the $g(\boldsymbol{x})$ are: $\partial g/\partial x^j = -2\gamma(x^j - y^j)$, $\partial^2 g/\partial x^{j2} = -2\gamma$, $\partial^m g/\partial x^{jm} = 0$, for $m \geq 3$, and $\frac{\partial^m g}{\partial x^1 \ldots \partial x^m} = 0$.

Applying the previous formula for $m = 1$ the first derivative is:

$$\begin{aligned}\frac{\partial}{\partial x^j} f(g(\boldsymbol{x})) &= \frac{\partial f}{\partial g(\boldsymbol{x})} \frac{\partial g}{\partial x^j} \\ &= f(g(\boldsymbol{x}))(-2\gamma(x^j - y^j)) \\ &= -2\gamma(x^j - y^j)k(\boldsymbol{x}, \boldsymbol{y}).\end{aligned}$$

The second derivative is:

$$\begin{aligned}\frac{\partial^2}{\partial x^{j2}} f(g(\boldsymbol{x})) &= \frac{\partial f}{\partial g(\boldsymbol{x})} \frac{\partial^2 g}{\partial x^{j2}} + \frac{\partial^2 f}{\partial g(\boldsymbol{x})^2} \left(\frac{\partial g}{\partial x^j}\right)^2 \\ &= f(g(\boldsymbol{x}))(-2\gamma) + f(g(\boldsymbol{x}))(4\gamma^2(x^j - y^j)^2) \\ &= 2\gamma(2\gamma(x^j - y^j)^2 - 1)k(\boldsymbol{x}, \boldsymbol{y}).\end{aligned}$$

The mixed derivative is:

$$\begin{aligned}\frac{\partial}{\partial x^j \partial x^i} f(g(\boldsymbol{x})) &= \frac{\partial f}{\partial g(\boldsymbol{x})} \frac{\partial^2 g}{\partial x^j \partial x^i} + \frac{\partial^2 f}{\partial g(\boldsymbol{x})^2} \left(\frac{\partial g}{\partial x^j}\right)\left(\frac{\partial g}{\partial x^i}\right) \\ &= f(g(\boldsymbol{x}))(0) + f(g(\boldsymbol{x}))(-2\gamma^2(x^j - y^j))(-2\gamma^2(x^i - y^i)) \\ &= 4\gamma^2(x^j - y^j)(x^i - y^i)k(\boldsymbol{x}, \boldsymbol{y}).\end{aligned}$$

## B Custom regression function

In this example we show the behaviour of the first and second derivatives for a multivariate input. A GP model is fitted over the dataset using the RBF kernel function. The experiment uses a custom linear multivariate function with two inputs, $x_1$ and $x_2$, as inputs:

$$y = ax_1 + bx_2, \tag{25}$$

where the coefficients $a$ and $b$ have varying values. Both $x_{1,2}$ were generated along the same range uniform distribution $\mathcal{U}([-20, 20])$ but there was a linear transformation $a = 5$, $b = 1$ from $([0, 20])$ and constant everywhere else, i.e. $a = b = 1$ from $([-20, 0])$.

The GP model smooths the piece-wise continuous function which results in some additional slopes than the original formulation. This is visible (see Fig 12) from the derivatives of the kernel model as the first derivative for the $x_1$ and $x_2$ components have positive values for the sensitivities of the slopes in the regions where $a$ and $b$ are equal to some constant, respectively. The second derivative for both $x_1$ and $x_2$ show the same effect except for curvature. This experiment successfully highlights the derivatives of the individual components as well as their combined sensitivity.

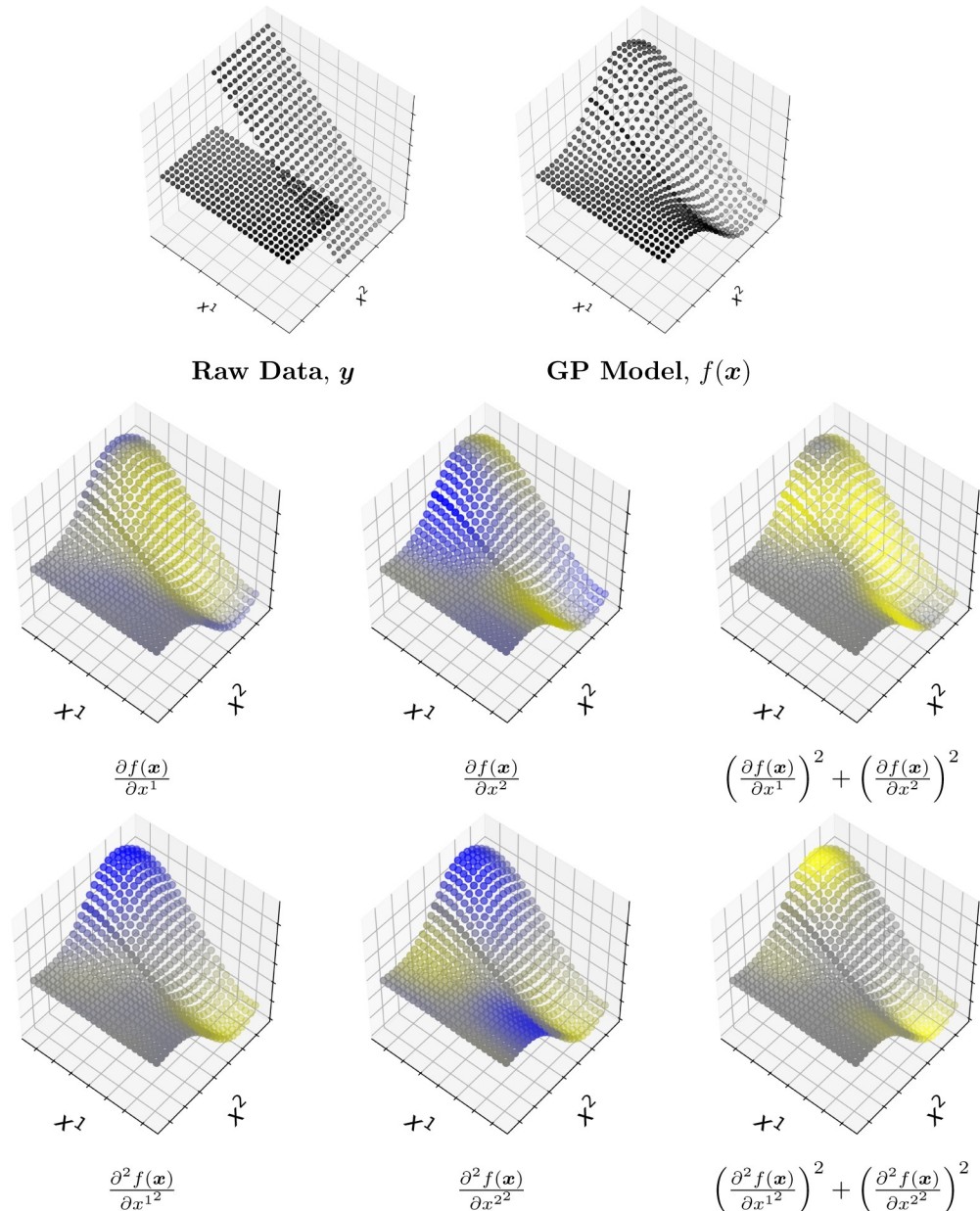

**Fig 12. First row**: The original toy data is displayed as well as the predicted GP model which presents a smoother curve. **Second row**: the first derivative in the $x_1, x_2$ direction and combined direction (the sensitivity) respectively. **Third row**: the second derivative in the $x_1$, $x_2$ direction and combined direction (the sensitivity) respectively. The yellow colored points represent the regions with positive values, the blue colored points represent the regions with negative values and the gray colored points represent the regions where the values are zero.

## Acknowledgments

J.E.J. thanks the European Space Agency (ESA) for support via the Early Adopter Call of the Earth System Data Lab project; M.D.M. thanks the ESA for the long-term support of this initiative.

## Author Contributions

**Conceptualization:** J. Emmanuel Johnson, Valero Laparra, Adrián Pérez-Suay, Miguel D. Mahecha, Gustau Camps-Valls.

**Data curation:** J. Emmanuel Johnson, Miguel D. Mahecha.

**Formal analysis:** J. Emmanuel Johnson, Valero Laparra, Adrián Pérez-Suay, Gustau Camps-Valls.

**Funding acquisition:** Gustau Camps-Valls.

**Investigation:** Valero Laparra, Adrián Pérez-Suay, Gustau Camps-Valls.

**Methodology:** J. Emmanuel Johnson.

**Resources:** J. Emmanuel Johnson, Miguel D. Mahecha, Gustau Camps-Valls.

**Software:** J. Emmanuel Johnson, Adrián Pérez-Suay.

**Supervision:** Valero Laparra, Miguel D. Mahecha, Gustau Camps-Valls.

**Validation:** J. Emmanuel Johnson, Valero Laparra, Adrián Pérez-Suay, Gustau Camps-Valls.

**Visualization:** Valero Laparra.

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
