## [Decision Letter · Decision Letter 0]

7 Feb 2020

PONE-D-19-17584

Kernel Methods and their Derivatives: Concept and Perspectives for the Earth System Sciences

PLOS ONE

Dear Mr Johnson,

Thank you for submitting your manuscript to PLOS ONE. After careful consideration, we feel that it has merit but does not fully meet PLOS ONE’s publication criteria as it currently stands. Therefore, we invite you to submit a revised version of the manuscript that addresses the points raised during the review process.

ACADEMIC EDITOR: 

Please revise the manuscript follow the reviewers' commentsPlease revise the manuscript in terms of languages and correct the typos

We would appreciate receiving your revised manuscript by Mar 15 2020 11:59PM. To enhance the reproducibility of your results, we recommend that if applicable you deposit your laboratory protocols in protocols.io, where a protocol can be assigned its own identifier (DOI) such that it can be cited independently in the future. For instructions see: http://journals.plos.org/plosone/s/submission-guidelines#loc-laboratory-protocols

We look forward to receiving your revised manuscript.

Kind regards,

Qichun 'Kit' Zhang, PhD

University of Bradford, UK

Journal Requirements:

2. Please explain the rationale for the development of your approach in light of recent research in this area, clearly indicating which problem with existing methods you are addressing.

3. Please clearly report at the beginning of your methods or results section which the key performance measures were to establish validity and utility of your method. Please also report clearly which statistical analysis was used to establish robustness of performance measures.

4. Please note that PLOS ONE requires that experiments, statistics, and other analyses must be performed to a high technical standard and described in sufficient detail to allow for reproducibility of the study (http://journals.plos.org/plosone/s/criteria-for-publication#loc-3). To demonstrate the performance of the method, we would expect comparisons to be drawn between existing state-of-the-art methods.

Additional Editor Comments:

This paper has been well-written with interesting topic. The reviewers return some critical comments, especially the novelty of the contribution as the kernel methods have been widely used. Moreover, some typos have to be corrected in the revised manuscript. Thus I believe that a major revision is necessary to improve the quality of the manuscript.

Reviewers' comments:

Reviewer's Responses to Questions

**Comments to the Author**

1. Is the manuscript technically sound, and do the data support the conclusions?

Reviewer #1: Yes

Reviewer #2: Partly

2. Has the statistical analysis been performed appropriately and rigorously? 

Reviewer #1: Yes

Reviewer #2: Yes

3. Have the authors made all data underlying the findings in their manuscript fully available?

Reviewer #1: Yes

Reviewer #2: Yes

4. Is the manuscript presented in an intelligible fashion and written in standard English?

Reviewer #1: Yes

Reviewer #2: Yes

5. Review Comments to the Author

Reviewer #1: I have read the paper, “Kernel Methods and their derivatives: Concept and

perspectives for the Earth system sciences”. The paper is well organized, well written and of high academic quality. I am happy to see that the authors have made a really good effort to apply the kernel methods and their derivatives for the Earth system sciences.

The majority of the work is very well, but there are some minor issues that I would recommend to address in the final version of the paper. These issues are provided as follows;

(1) The citation of Figure is not consistent. Cite the figure as (Fig. xx) or (Figure xx).

(2) At line # 216 citation is missing

(3) At line # 61 what mean of m-th order? As it is not defined before.

(4) At line # 83 change the text, Section 7.4 with Section 6.

(5) The detail is missing in conclusion section, What was done in this paper related to Earth system sciences?

Reviewer #2: Manuscript Draft: PONE-D-19-17584 - Kernel Methods and their Derivatives: Concept and Perspectives for the Earth System Sciences

I. Summary

This article surveys the application of kernel methods to several supervised and unsupervised machine learning tasks, including Gaussian processes for regression, SVMs for classification, density estimation, and dependence estimation, with specific focus on showing the potentials of using the derivatives of kernel functions to gain insight of the function learned and to design kernel machines. This study claims the following contributions:

• To show a way to interpret the functions learned by various kernel methods by using their derivatives;

• To provide the explicit analytic form of the 1st and 2nd derivatives of some kernel functions and generic formulas to compute arbitrary order derivatives;

• To provide illustrative toy examples as well as a real-world problem with earth system data to demonstrate the usefulness of the derivatives.

While the topic of the article is interesting and the paper is well organized, I don’t think the current version of the article is ready for publication because the paper provides neither a synergistic framework for better understanding the kernel methods nor sufficient evidences to demonstrate their effectiveness in earth system science, failing to well justify the claimed contribution. I provide more specific comments in the following section.

II. Major issues

1. As the derivations of kernel functions’ derivatives are not new, one key theoretical contribution of this paper is to give more insights into the supervised and unsupervised kernel learning methods by relating the derivatives to the associated characteristic information, such as margin, sensitivity, and leverage. However, the new development of the present article is not clear to me compared to the previous study, i.e. [28,29,51,30], instead of summarizing the findings in these papers and running new toy examples. To make the article clearer, can the authors provide more details and discussion about the differences compared to the above mentioned study?

2. The definition 1 and Theorem 1 in Page 5 seem not relevant to the main material. Are they serve any purpose?

3. Following last comment, the kernel classification in Section 4 is not used in the earth system problem.

4. Section 7 looks interesting. But it is too short without sufficient evidences to demonstrate the effectiveness of kernel methods. Besides, the readers may want to know the background of spatio-temporal earth system data and the state-of-art of kernel methods in this domain, but this information is missing in the article.

5. What does the color bar in Figure 7 mean? Also, the font size of the numeric values in the plots is too small.

6. What are the values of a and b to generate the data in Figure 10? There are typos in the caption.

III. Minor issues

1. In Abstract, the phrase “…various kernel methods in a much more intuitively that commonly assumed…” is strange and should be reworded.

2. In the last paragraph of Introduction, the sentence “Section 7.4 pays attention to …” seems to have a wrong reference to the section number. Please double check.

3. Please double check the symbol in Eq. (9)

4. The reference is missing in Section 3.1.

5. In the caption of Figure 2, it seems the regularized and unregularized cases are wrongly referred.

6. A symbol error in Line 260, Page 11

7. In the caption of Figure 3, the phrase “The predicted has red and green…” has typos. Typos are also found in Figure 10. Please double check the spelling and grammar in all the captions to avoid unnecessary confusion to the readers.

8. What are the u, v in Line 494, Page 19.

There are more typos in the paper, I encourage the authors check the writing carefully in the revised version.

6. PLOS authors have the option to publish the peer review history of their article (what does this mean?). If published, this will include your full peer review and any attached files.

Reviewer #1: No

Reviewer #2: No

---

## [Author Response · Author response to Decision Letter 0]

6 Apr 2020

We have attached a response letter as a separate PDF. Copy Pasted the response below.

Reviewer I

Summary

I have read the paper, “Kernel Methods and their derivatives: Concept and perspectives for the Earth system sciences”. The paper is well organized, well written and of high academic quality. I am happy to see that the authors have made a really good effort to apply the kernel methods and their derivatives for the Earth system sciences. The majority of the work is very well, but there are some minor issues that I would recommend to address in the final version of the paper.

Thank you for the positive comments. To the authors’ knowledge, this is the first time that the derivatives of kernel methods have been showcased and summarized in such a detailed matter, collectively, and for the most widely used kernel machines and learning paradigms. In addition, we have applied these methods to toy examples, which highlight understanding, and to example applications in the very relevant field of Earth sciences. We have performed a thorough revision of the language and corrected the typos as suggested. We feel that with the additional changes, the manuscript has improved and is ready for your consideration.

Comments

Minor Issues

The citation of Figure is not consistent. Cite the figure as (Fig. xx) or (Figure xx).

At line # 61 what mean of m-th order? As it is not defined before.

At line # 83 change the text, Section 7.4 with Section 6

At line # 216 citation is missing

 All of the minor issues and typos have been corrected. 

The detail is missing in conclusion section, What was done in this paper related to Earth system sciences?

Following the reviewers’ suggestions, we have updated the conclusion section by emphasizing our contributions to the field of Earth sciences too. We have added the following paragraphs:

“To recap, we have provided intuitive explanations for derivatives of each kernel method through illustrative toy examples, and also highlighted the links between each of the formulations with concise expressions to showcase the similarities. We show that 1) the derivatives of kernel regression models (such as GPs) allows one to do sensitivity analysis to find relevant input features, 2) the derivatives of kernel classification models (such as SVMs) also allows one to do sensitivity analysis and relate it to the margin, 3) the derivatives of kernel density estimators (KDE) allows one to describe the ridge of the estimated multivariate densities, 4) the derivatives of kernel dependence measures (such as HSIC) allows one to visualize the magnitude and change of direction in the dependencies between two multivariate datasets. We have also given proof-of-concept examples of how they can be used in challenging applications with spatial-temporal Earth datasets. In particular, we show that 1) we can express the spatial-temporal relationships as inputs to regression algorithms and evaluate their relevance for prediction of essential climate variables, 2) we can assess the margin for classification models in drought detection, as a way to identify the most sensitive points/regions for detection, 3) the ridges can be used as indicators of potential regions of interest due to their location in the PDF, which could be related to anomalies, and 4) we can detect changes in dependence between two events during an extreme heatwave event.”

Reviewer II

Summary

This article surveys the application of kernel methods to several supervised and unsupervised machine learning tasks, including Gaussian processes for regression, SVMs for classification, density estimation, and dependence estimation, with specific focus on showing the potentials of using the derivatives of kernel functions to gain insight of the function learned and to design kernel machines. This study claims the following contributions:

• To show a way to interpret the functions learned by various kernel methods by using their derivatives;

• To provide the explicit analytic form of the 1st and 2nd derivatives of some kernel functions and generic formulas to compute arbitrary order derivatives;

• To provide illustrative toy examples as well as a real-world problem with earth system data to demonstrate the usefulness of the derivatives.

While the topic of the article is interesting and the paper is well organized, I don’t think the current version of the article is ready for publication because the paper provides neither a synergistic framework for better understanding the kernel methods nor sufficient evidences to demonstrate their effectiveness in earth system science, failing to well justify the claimed contribution. There are more typos in the paper. I encourage the authors’ check the writing carefully in the revised version.

Thank you for the comments. We want to clarify the following:

To authors’ knowledge, this is the first time the derivatives of kernel methods have been showcased and summarized in such a detailed matter: there have been no collective studies for the most widely used kernel machines or learning paradigms with illustrative toy examples. We have included four key learning paradigms in this study: classification, regression, density and dependence estimation. We have highlighted this contribution more clearly, and outlined the relations of derivatives of kernel methods with interesting features of the learned functions. For instance, we showed that the derivatives of kernel regression are useful for sensitivity analysis, derivatives of the SVM classification function are related to the margin, derivatives can be used to describe the ridge in densities, while derivatives of kernel dependence measures allow visualization and optimization of the measure.

Another novelty of our work is that we show the potential of derivatives of kernel methods in real, challenging problems. We are unaware of any collective study showing examples with real-world data, and in our case, it complements the toy examples. We demonstrated the use of the derivative in several applications involving Earth data cubes: how one can gain knowledge about the spatio-temporal relations in estimation settings, how to interpret classifiers trained for detecting extreme events, how to estimate the “interestingness” of spatial regions by looking at the joint PDF estimation of temperature and moisture data cubes, and how to detect changes in dependence by characterizing the derivatives of kernel dependence measures between Earth variables. Implications in the field of Earth sciences are obvious.

Following the reviewer's comment, we have now included a new experiment on the use of derivatives of SVM classification of the Russian heatwave to complete the 1-to-1 correspondence between the toy examples and the applications. Thanks for the suggestion.

In addition, and following the reviewer's suggestions, we have performed a thorough revision of the language use and corrected typos. 

We feel that, with the included changes, the manuscript has improved substantially and is ready for your consideration.

Major Issues

As the derivations of kernel functions’ derivatives are not new, one key theoretical contribution of this paper is to give more insights into the supervised and unsupervised kernel learning methods by relating the derivatives to the associated characteristic information, such as margin, sensitivity, and leverage. However, the new development of the present article is not clear to me compared to the previous study, i.e. [28,29,51,30], instead of summarizing the findings in these papers and running new toy examples. To make the article clearer, can the authors provide more details and discussion about the differences compared to the above mentioned study?

Thank you for raising this point. We partially agree with the reviewer: while the kernel derivatives themselves are not new, their properties and their use in general kernel methods have not been reported collectively and extensively so far in the literature. We present the first paper that exclusively focuses on kernel derivatives in general from theory to practice, and shown both in simulated and complex settings. In each learning paradigm, we have identified scattered concepts in the literature: derivatives of kernel methods connect dots between distant concepts such as leverage scores, classifier margin, sensitivity analysis or densities ridges/principal curves. Many of these concepts were never treated collectively under the same umbrella, and we have expressed all of these aspects in our proposed paper. 

The paper is not a mere collection of formulations; it is of practical interest with important implications in the Earth sciences. This is why we have not only provided toy illustrative examples, but also relevant applications on spatio-temporal Earth data. The latter is aimed to inspire applications in the same domain and other domains with similar data characteristics, e.g. signal processing (reference 1) and remote sensing (reference 2). We have highlighted all of these contributions in the introduction and conclusions sections, as well as included a summarizing table of the formulations and intuitions under the kernel derivatives for all methods {sensitivity, margin, ridge, leverage} at the end of the methods section. 

References:

Rojo‐Álvarez et. al. - Digital Signal Processing with Kernel Methods - 2018

Camps-Valls & Bruzzone - Kernel Methods for Remote Sensing Data Analysis - 2009

The definition 1 and Theorem 1 in Page 5 seem not relevant to the main material. Are they serve any purpose?

This is a good point and we agree that this background kernel literature would be in principle unnecessary in a standard kernel methods paper. Nevertheless, we included it merely to build on the properties of kernel methods that allow to express the derivatives of the implemented function f(x) as a linear expansion of derivatives of the kernel function K(x,x'). This point, we feel, is strictly necessary for the general reader, not for experts in kernels (note that the paper will be of interest to Earth scientists too). However, we realize that it is not for all readers, so we have included a sentence at the beginning of the paragraph highlighting the main purpose of the section, and also concluded the paragraph with specific references PDF and PDF for any interested reader who would like to go further into the kernel literature.

Following last comment, the kernel classification in Section 4 is not used in the earth system problem.

Thank you for pointing this out. In the new version of the manuscript we have included a real classification experiment focused on anomaly (heatwave) classification, for which we show the usefulness of derivatives and their relation to the margin. We agree that this completes the paper nicely so we have now 1-to-1 correspondence between toy examples and ESDC applications. Thanks for the suggestion.

Section 7 looks interesting. But it is too short without sufficient evidences to demonstrate the effectiveness of kernel methods. Besides, the readers may want to know the background of spatio-temporal earth system data and the state-of-art of kernel methods in this domain, but this information is missing in the article.

Thank you for your comment. This section was included to illustrate the utility of the derivatives of kernel methods in real scenarios. The section does not pretend to be a review of kernel methods in Earth sciences, see this book for an overview of the field. Broadly speaking, kernel machines have been used in many problems in remote sensing and geosciences; from estimation of biophysical parameters to classification and anomaly detection, as well as to cluster regions and events, and to estimate dependencies between variables. However, the use of the derivatives of kernel machines are restricted to few examples, and we feel they can (and should) be exploited further, especially in spatio-temporal Earth data cubes, which is a missing part at the moment in the literature. 

The selection of the spatio-temporal data for this section gives a very good opportunity to evaluate our proposal in a unique and challenging setting. Note that very few kernel methods have been used in such settings (restricted to new forms of KPCA and multi-temporal change detection with SVMs), and that the derivatives have never been exemplified before. This is why we decided to use this opportunity to do a proof-of-concept with our overview paper, and this section is essentially aimed at illustrating the use of derivatives on real, multivariate, nonlinear spatio-temporal Earth data; a challenging and important problem setting. Examples in this dataset are also illustrative and could motivate the use and adoption in other domains and/or similar problems sharing similar data characteristics, such as in neuroscience, signal processing and communications or remote sensing.

Following the reviewer’s suggestions, we have added several changes. First, we have included more contextualization about the use of kernel methods for Earth sciences at the introduction, some key references in the field, and motivated the use of kernel methods for spatio-temporal multivariate data in general and the Earth sciences in particular. We also added the new section on derivatives of kernel methods in anomaly detection in this setting using SVMs, and relating them to the concept of margin. Finally, in the conclusions section, we have included more discussion about the implications in the geosciences, as well as hypothesized on the potential use in other applications and/or domains in science and engineering. 

What does the color bar in Figure 7 mean? Also, the font size of the numeric values in the plots is too small.

The colorbar represents the sensitivity; we have indicated now in the new version of the manuscript. The range is colored appropriately per variable. The font size was indeed too small; we have increased it for better visualization. Thanks.

What are the values of a and b to generate the data in Figure 10? There are typos in the caption.

Thanks for pointing out these minor issues. We have corrected the typos and added some description of the generated data in the very same caption. Thanks.

Minor Issues

In Abstract, the phrase “…various kernel methods in a much more intuitively that commonly assumed…” is strange and should be reworded.

In the last paragraph of Introduction, the sentence “Section 7.4 pays attention to …” seems to have a wrong reference to the section number. Please double check.

Please double check the symbol in Eq. (9)

The reference is missing in Section 3.1.

In the caption of Figure 2, it seems the regularized and unregularized cases are wrongly referred.

A symbol error in Line 260, Page 11

In the caption of Figure 3, the phrase “The predicted has red and green…” has typos. Typos are also found in Figure 10. Please double check the spelling and grammar in all the captions to avoid unnecessary confusion to the readers.

What are the u, v in Line 494, Page 19.

Thanks for the thorough revision. We have corrected all the typos and revised the language usage and grammar exhaustively.

---

## [Decision Letter · Decision Letter 1]

25 Jun 2020

Kernel Methods and their Derivatives: Concept and Perspectives for the Earth System Sciences

PONE-D-19-17584R1

Dear Dr. Johnson,

We’re pleased to inform you that your manuscript has been judged scientifically suitable for publication and will be formally accepted for publication once it meets all outstanding technical requirements.

Kind regards,

Qichun 'Kit' Zhang, PhD

University of Bradford, UK

Additional Editor Comments (optional):

Two reviewers have returned the reports which show that the revised manuscript has been improved strongly in terms of contribution and quality. Therefore, I recommend to accept this paper after correcting the typos.

Reviewers' comments:

Reviewer's Responses to Questions

**Comments to the Author**

1. If the authors have adequately addressed your comments raised in a previous round of review and you feel that this manuscript is now acceptable for publication, you may indicate that here to bypass the “Comments to the Author” section, enter your conflict of interest statement in the “Confidential to Editor” section, and submit your "Accept" recommendation.

Reviewer #1: All comments have been addressed

Reviewer #2: All comments have been addressed

2. Is the manuscript technically sound, and do the data support the conclusions?

Reviewer #1: Yes

Reviewer #2: Yes

3. Has the statistical analysis been performed appropriately and rigorously? 

Reviewer #1: Yes

Reviewer #2: Yes

4. Have the authors made all data underlying the findings in their manuscript fully available?

Reviewer #1: Yes

Reviewer #2: Yes

5. Is the manuscript presented in an intelligible fashion and written in standard English?

Reviewer #1: Yes

Reviewer #2: Yes

6. Review Comments to the Author

Reviewer #1: (No Response)

Reviewer #2: The manuscript was well revised and Reviewer's comments have been well addressed.

Please note that there are still few typos about the inconsistent notations: e.g., in the last paragraph of Introduction, "section 8" is not capitalized.

7. PLOS authors have the option to publish the peer review history of their article (what does this mean?). If published, this will include your full peer review and any attached files.

Reviewer #1: No

Reviewer #2: No

---

## [Editor Report · Acceptance letter]

17 Jul 2020

PONE-D-19-17584R1 

Kernel Methods and their Derivatives: Concept and Perspectives for the Earth System Sciences 

Dear Dr. Johnson:

I'm pleased to inform you that your manuscript has been deemed suitable for publication in PLOS ONE. Congratulations! Your manuscript is now with our production department. 

Kind regards, 

on behalf of

Dr. Qichun Zhang 

Academic Editor

PLOS ONE